# AMPAR/TARP stoichiometry differentially modulates channel properties

**Federico Miguez-Cabello[1], Nuria Sánchez-Fernández[1], Natalia Yefimenko[1], Xavier Gasull[1,2], Esther Gratacòs-Batlle[1], David Soto[1,2]***

[1]Laboratori de Neurofisiologia, Departament de Biomedicina, Facultat de Medicina i Ciències de la Salut, Institut de Neurociències, Universitat de Barcelona, Barcelona, Spain; [2]Institut d'Investigacions Biomèdiques August Pi i Sunyer (IDIBAPS), Barcelona, Spain

**Abstract** AMPARs control fast synaptic communication between neurons and their function relies on auxiliary subunits, which importantly modulate channel properties. Although it has been suggested that AMPARs can bind to TARPs with variable stoichiometry, little is known about the effect that this stoichiometry exerts on certain AMPAR properties. Here we have found that AMPARs show a clear stoichiometry-dependent modulation by the prototypical TARP $\gamma 2$ although the receptor still needs to be fully saturated with $\gamma 2$ to show some typical TARP-induced characteristics (i.e. an increase in channel conductance). We also uncovered important differences in the stoichiometric modulation between calcium-permeable and calcium-impermeable AMPARs. Moreover, in heteromeric AMPARs, $\gamma 2$ positioning in the complex is important to exert certain TARP-dependent features. Finally, by comparing data from recombinant receptors with endogenous AMPAR currents from mouse cerebellar granule cells, we have determined a likely presence of two $\gamma 2$ molecules at somatic receptors in this cell type.

## Introduction

Glutamate is a crucial neurotransmitter in the central nervous system (CNS), mediating the vast majority of the fast-excitatory synaptic transmission acting on postsynaptic ionotropic glutamate receptors. Among these, α-amino-3-hydroxy-5-methyl-4-isoxazolepropionic acid (AMPA) receptors (AMPARs) are fundamental players in the synaptic course besides their pivotal role as regulators of synaptic plasticity. AMPARs are homo or heterotetrameric structures composed of four GluA subunits (GluA1-4) (*Traynelis et al., 2010*) and their biophysical properties are dramatically changed depending on the subunits that conform the ion channel, with the presence or absence of the GluA2 subunit a critical determinant of channel behaviour. In particular, while GluA2-lacking AMPARs are permeable to both $Na^+$ and $Ca^{2+}$ ions, AMPARs containing the GluA2 subunit are impermeable to $Ca^{2+}$. GluA2-containing AMPARs do not allow $Ca^{2+}$ influx through the channel due to an RNA editing process that results in the replacement of a neutral glutamine for the positive amino acid arginine in the cation selectivity filter region of this subunit at the so-called Q/R site (*Hollmann et al., 1991*; *Burnashev et al., 1992*). GluA2-exclusive RNA editing process occurs in 99% of native GluA2 subunits (*Geiger et al., 1995*; *Kawahara et al., 2003*). This change not only affects $Ca^{2+}$ permeability but also single-channel conductance, which is decreased in GluA2-containing AMPARs mainly due to the lack of $Ca^{2+}$ permeability (*Swanson et al., 1997*). Editing at the Q/R site also impairs the blocking effect of endogenous polyamines on these receptors at depolarized membrane potentials compared to GluA2-lacking AMPARs, which are strongly blocked by spermine (*Bowie and Mayer, 1995*; *Kamboj et al., 1995*; *Koh et al., 1995*). Thus, AMPARs are frequently classified as $Ca^{2+}$-impermeable *vs.* $Ca^{2+}$-permeable (CI *vs.* CP-AMPARs – or GluA2-containing *vs.* GluA2-lacking). Finally, the Q/R editing of the GluA2 subunit powerfully influences AMPAR tetramerization and

**\*For correspondence:**
davidsoto@ub.edu

**Competing interests:** The authors declare that no competing interests exist.

strongly disfavours formation of GluA2 homotetramers (*Greger et al., 2003*) although a marginal population of GluA2 homomers (~1%) have been found to reach the plasma membrane in vivo (*Zhao et al., 2019*).

While AMPAR gating – and also trafficking – properties are determined by their subunit composition, these features are also strongly dependent on AMPAR-associated transmembrane proteins that behave as auxiliary subunits of the receptor. In the last 15 years, the number of interacting proteins that can act as modulatory partners of AMPARs has vastly increased. Stargazin and other TARPs (*Transmembrane AMPAR Regulatory Proteins*) were the first AMPAR-modulating proteins to be discovered (*Chen et al., 2000*; *Tomita et al., 2003*). TARPs are one of the most studied AMPAR auxiliary proteins because of their indispensable role in neuronal physiology (*see Payne, 2008* for *review*) and in different types of synaptic plasticity (*Rouach et al., 2005*; *Louros et al., 2014*; *Sullivan et al., 2017*; *Louros et al., 2018*). It is well-known that members of the TARP family can modify biophysical properties of AMPARs by increasing conductance, slowing down kinetics or diminishing polyamine block in CP-AMPARs (*Straub and Tomita, 2012*; *Haering et al., 2014*; *Jackson and Nicoll, 2011*; *Greger et al., 2017*). Furthermore, TARPs can modify AMPAR pharmacology such as kainate-evoked responses (*Turetsky et al., 2005*; *Kott et al., 2007*). However, much less is known about the number of TARP molecules that can be present in an AMPAR complex although AMPAR/TARP stoichiometry has been investigated by single-molecule subunit counting (*Hastie et al., 2013*) and electrophysiological studies, which demonstrated that AMPAR efficiency to kainate varies depending on the number of TARPs present in the receptor (*Shi et al., 2009*). More recently, functional studies together with high definition structural data provided evidence for a favoured 2-TARP per AMPAR stoichiometry (*Dawe et al., 2019*; *Herguedas et al., 2019*). However, with only a small number of studies investigating AMPAR/TARP stoichiometry, it is still not established whether TARPs can modulate AMPARs in a stoichiometry dependent manner.

In the present work we have determined how the number of γ2 (the prototypical TARP) per AMPAR complex modifies the biophysical properties of these receptors in a stoichiometry dependent manner in both CP and CI-AMPARs. To approach this issue, we have used AMPAR/TARP fusion proteins to fix stoichiometries of 2 and 4 TARPs per AMPAR. Our results show a complex stoichiometry-dependent modulation with important differences observed between CP and CI-AMPARs. Moreover, we have attempted to reveal the endogenous functional AMPAR/TARP stoichiometry in cerebellar granule cells (CGCs) by correlating results obtained in recombinant AMPARs with recordings on CGC cultures. We have taken advantage that this cerebellar neuronal type expresses a limited variety of GluA and TARP subunits together with a lack of cornichons expression on the plasma membrane (another important AMPAR auxiliary protein) (*Schwenk et al., 2009*; *Shi et al., 2010*). We propose that just 2 molecules of the TARP family (specifically γ2) determine functional somatic AMPAR properties in CGCs acting in a complex manner on both GluA2 and GluA4c subunits.

## Results

### γ2 induce a graded change in CP-AMPAR kinetics

The archetypal TARP γ2 slows deactivation and desensitization of AMPAR-mediated responses (*Priel et al., 2005*), as observed with the other members of the TARP family (*Soto et al., 2009*). However, the potential impact of a different TARP stoichiometry on AMPAR kinetics is quite unknown. We wondered whether an increasing number of TARP subunits present into the AMPAR complex modulates channel properties in a graded way or whether the presence of two TARP molecules is enough to provide AMPARs with a TARPed behaviour. We studied two fixed stoichiometries using the fusion protein GluA1:γ2, which comprises GluA1 and γ2 in the same protein product that has been previously validated (*Soto et al., 2014*). We co-transfected tsA201 cells with GluA1 and/or GluA1:γ2 in such a way that GluA1 homomeric CP-AMPARs had zero, putatively two or four TARPs per receptor. Then we extracted outside-out patches from transfected cells, and we applied a 10 mM glutamate step during 100 ms with a piezoelectric controller to acquire fast AMPAR-mediated responses.

We analysed desensitization kinetics of GluA1 homomeric receptors in these three conditions. Desensitization of GluA1 in the presence of the saturating concentration of glutamate – measured as the weighted time constant ($\tau_w$) – showed a clear TARP dependence with an increase in

desensitization time as the number of γ2 units in the complex increased (2.32 ± 0.16 ms, 3.77 ± 0.39 ms and 6.70 ± 0.41 ms for 0-TARPs, 2-TARPs and 4-TARPs respectively; p<0.05 for comparisons between all groups; one-way ANOVA; *Figure 1A–B*).

In contrast to the well-defined step changes observed for desensitization kinetics, there was not a crystal-clear graded change in steady state current (*Figure 1C*). A significant change was detected when the CP-AMPAR was fully saturated with TARPs (2.78 ± 1.04% for 0-TARPs *vs.* 13.95 ± 1.85%

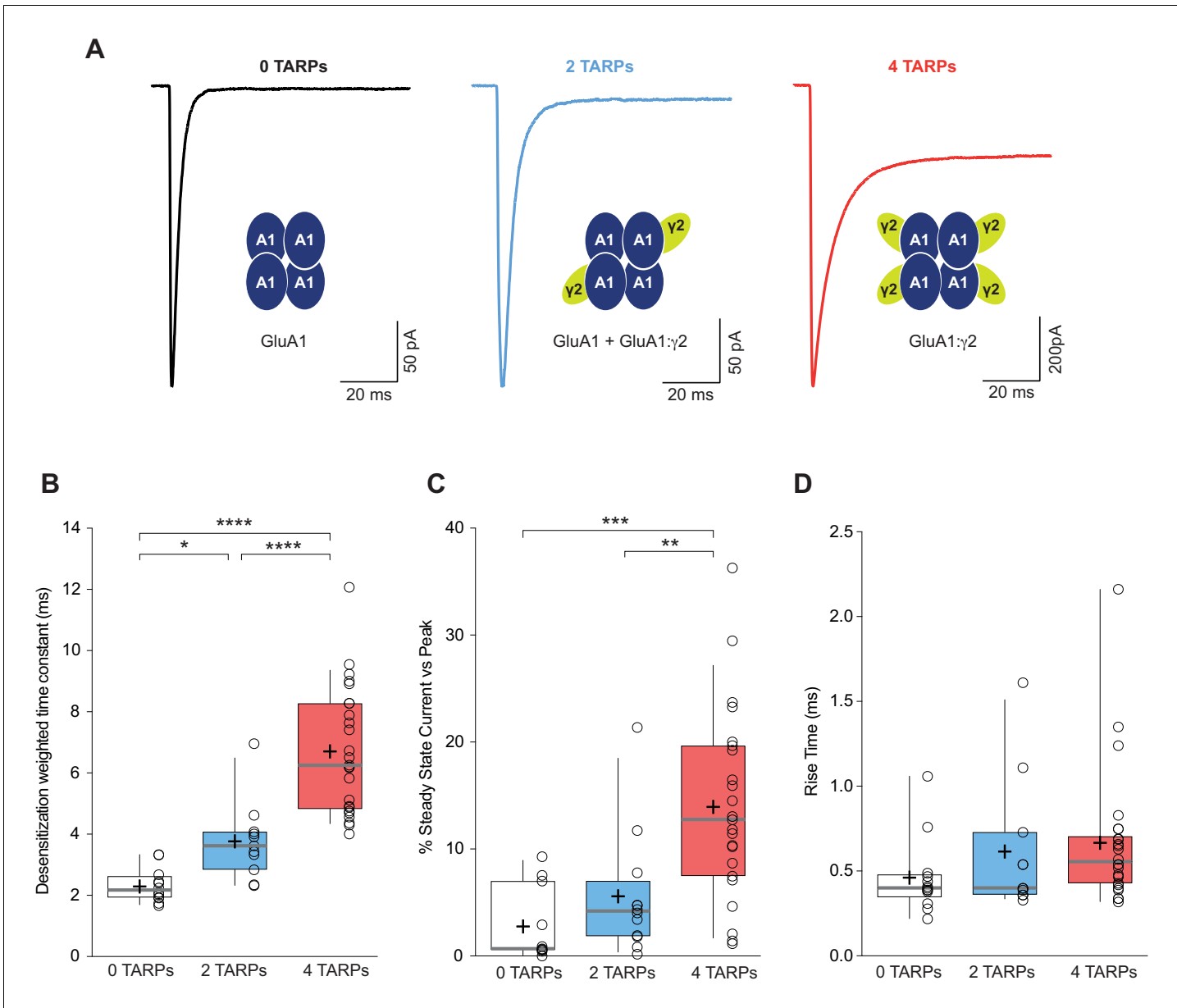

**Figure 1.** CP-AMPAR kinetics are differentially affected by AMPAR-TARP stoichiometry. (**A**) Traces evoked at −60 mV by rapid application of 10 mM glutamate to outside-out patches from cells expressing GluA1 alone (black; average of 39 responses) or together with 2 (blue; average of 37 responses) or 4 (red; average of 67 responses) γ2 subunits. (**B**) Pooled data of the weighted time constant of desensitization (τ$_{w, des}$). Box-and-whisker plots indicate the median value (gray line), the 25–75th percentiles (box), and the 10–90th percentiles (whiskers); crosses and open circles represent mean and the individual experimental values respectively. (**C**) Pooled data showing the increase in the steady state current only in 4 TARPed CP-AMPARs. (**D**) Rise time of glutamate-activated currents is not affected by TARPs. The data from this figure containing statistical tests applied, exact sample number, p values and details of replicates are available in '*Figure 1—source data 1*'.

The online version of this article includes the following source data for figure 1:

**Source data 1.** Kinetic properties of CP-AMPARs.

for 4-TARPs; p<0.001; one-way ANOVA) although a graded variation cannot be discarded since 2-TARPs and 4-TARPs conditions significantly differed (5.58 ± 1.70% for 2-TARPs *vs.* 13.95 ± 1.85% for 4-TARPs; p<0.01).

We next analysed the kinetics of the current activation (rise time) and we did not observe a significant increase in the time to reach the peak current (0.46 ± 0.06 ms, 0.61 ± 0.12 ms and 0.67 ± 0.08 ms for 0-TARPs, 2-TARPs and 4-TARPs respectively; one-way ANOVA; p>0.05 for all comparisons between groups; *Figure 1D*).

TARPs also speed the recovery from desensitization of AMPARs (*Priel et al., 2005*; *Gill et al., 2012*; *Cais et al., 2014*; *Carbone and Plested, 2016*) so we checked if this phenomenon was stoichiometry dependent. We applied paired pulses of glutamate separated by 20 to 720 ms intervals onto patches from cells expressing GluA1, GluA1+GluA1:γ2 or GluA1:γ2. *Figure 2A* shows typical recordings for the three conditions mentioned above. We then calculated the desensitization recovery rate and we observed a graded effect, with the 2-TARPs condition halfway between the slow recovery of 0-TARPs and the quicker recovery of 4-TARPs (*Figure 2B*). Specifically, we found time constants (τ) of 98.57 ± 7.35 ms for 0-TARPs, 68.91 ± 5.92 ms for 2-TARPs and 53.86 ± 4.78 ms for 4-TARPs (*Figure 2C*; n = 9, 14 and 10 respectively). Despite the seemingly graded effect due to a variable stoichiometry, we did not find differences between the 2-TARP and 4-TARP conditions (p>0.05; one-way ANOVA).

## CP-AMPAR polyamine block attenuation strongly depends on γ2 dosage

An important canonical property of CP-AMPARs is the strong intracellular polyamine block of the channel especially at depolarized potentials, which translates into a characteristic inwardly rectifying current-voltage relationship (*Kamboj et al., 1995*). This strong block by polyamines is attenuated as a consequence of TARP modulation (*Soto et al., 2007*; *Soto et al., 2009*). Thus, we investigated if this weakening of the block followed a stepwise pattern similar to the one observed for desensitization kinetics. A strong dependence on the number of γ2 molecules associated with the CP-AMPAR in such attenuation across different membrane voltages was observed (*Figure 3A–B*). The rectification index (RI; +60 mV /-60 mV; *Figure 3C*) clearly demonstrates a stoichiometry dependence of γ2-mediated attenuation of spermine block (0.056 ± 0.004 for 0-TARPs; 0.128 ± 0.011 for 2-TARPs and 0.274 ± 0.021 for 4-TARPs; p<0.05 for comparisons between all groups; one-way ANOVA).

The results obtained in *Figures 1–3* supported the idea of a graded modulation of GluA1 homomeric receptors depending on the number of TARPs. However, the possibility existed that in the studied 2-TARPed condition there was a mixture of two distinct populations of AMPARs (0-TARPed and 4-TARPed) rather than a pure population of heteromeric GluA1-GluA1:γ2. This would account for the intermediate phenotype observed in most of the parameters analysed. Therefore, we decided to transfect tsA201 cells with GluA1(Q) together with its edited variant GluA1(R) and use the polyamine block as an indicator of the existence of an heteromeric population. GluA1(Q) homomers are strongly blocked by polyamines and since GluA1(R) homomers are strongly disfavoured, a linear response would be indicative of a GluA1(Q)-GluA1(R) heteromeric receptor. We set the RI cut point as 0.7, considering a lower value in the responses as evidence for a substantial contamination with GluA1(Q) homomers. We found that 80% of the responses showed a RI >0.7 (8 out of 10 recordings). When we co-transfected GluA1(Q):γ2 with GluA1(R) we observed RIs above 0.7 in all recorded patches (n = 9), a percentage that did not differ from the TARPless condition (p=0.4737; Fisher's exact test). We examined the desensitization kinetics of these recordings to confirm the effect of γ2 in the complex. Importantly, as detected with GluA1(Q) forms, in the GluA1(R)-containing receptors the weighted time constant (τ_w) of the TARPed receptor was slowed significantly compared with the TARPless receptor (2.87 ± 0.47 for 0-TARPs *vs.* 4.32 ± 0.46 for 2-TARPs; p<0.05; student's *t*-test; data not shown). This suggests that the results obtained in the 2-TARPed conditions with GluA1(Q) from *Figures 1–3* were acquired putatively from a major 2-TARPed population, although it cannot be completely ruled out the presence of a small amount of 'contaminating' homomeric AMPARs.

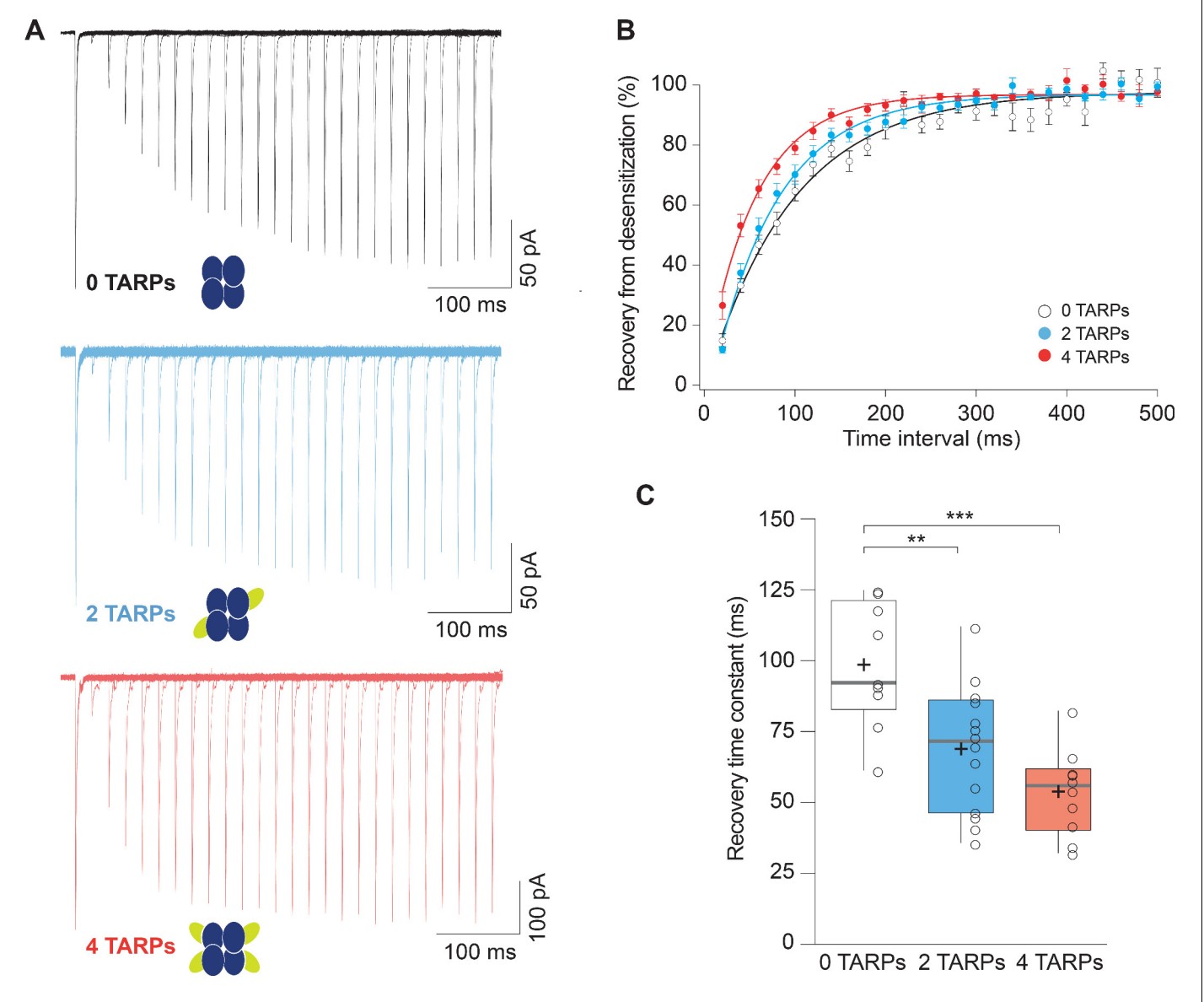

**Figure 2.** Recovery from desensitization of CP-AMPARs is enhanced in a graded manner with increased γ2. (**A**) Representative traces of a two-pulse protocol with increasing time interval between pulses for CP-AMPAR without γ2 TARP (GluA1 homomers; black), with 2 γ2 TARPs (blue) and with 4 γ2 TARPs (red). (**B**) Recovery from desensitization dynamics where it can be observed a gradual diminishment in the time needed to recover as the number of γ2 increases. (**C**) Recovery time constant values for the experiments showed in A and B. The data from this figure containing statistical tests applied, exact sample number, p values and details of replicates are available in '*Figure 2—source data 1*'.
The online version of this article includes the following source data for figure 2:

**Source data 1.** Recovery from desensitization of CP-AMPARs.

## CP-AMPAR conductance increase by γ2 needs a fully saturated receptor

We next performed non-stationary fluctuation analysis (NSFA) to obtain single-channel conductance and peak open probability from these macroscopic responses. *Figure 4A* shows representative responses from TARPless, 2 TARPed and 4 TARPed AMPARs together with their corresponding NSFA (*Figure 4B*). We obtained conductance values for TARPless GluA1 homomeric receptors (16.58 ± 0.69 pS; *Figure 4C*) similar to conductance values of 16.53 pS we previously described in the laboratory (*Gratacòs-Batlle et al., 2014*). As expected GluA1:γ2 fusion protein (4-TARPed) responses were increased ~1.4–1.5 fold as previously described when GluA1 is co-transfected

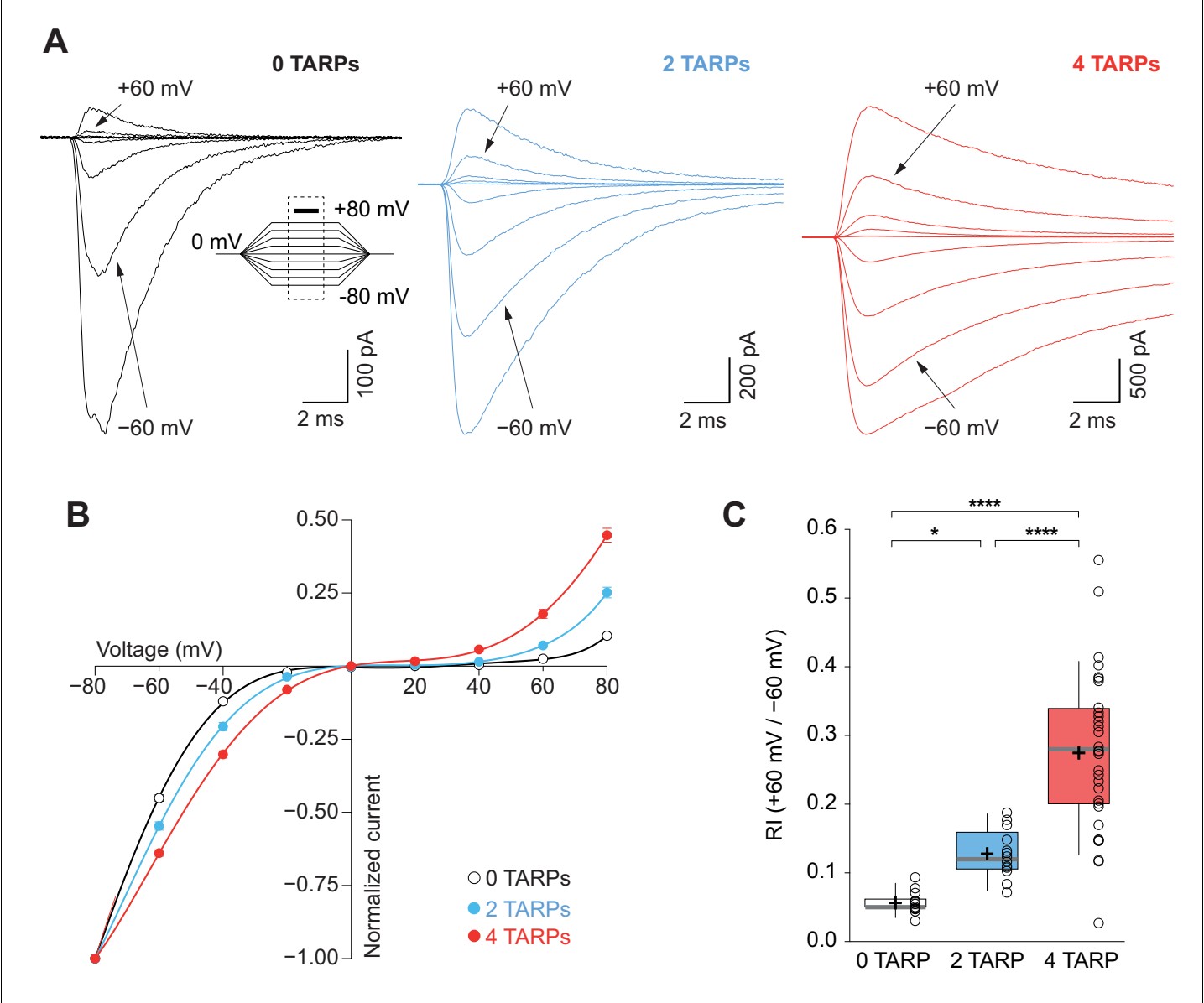

**Figure 3.** CP-AMPAR polyamine block attenuation is dependent on TARP dosage. (**A**) Representative glutamate-evoked currents from outside-out patches at different membrane potentials from −80 to +80 in 20 mV increments from cells expressing CP-AMPARs, GluA1 (black), GluA1+GluA1: γ2 (blue) and GluA1: γ2 (red). Bottom: Traces at +60 mV and −60 mV membrane potentials are marked. (**B**) I-V relationships constructed from glutamate-evoked peak currents of patches held at different membrane potentials in different AMPAR-TARP stoichiometries. (**C**) Pooled data showing an increase in the RI as the number of TARPs per CP-AMPAR increases. The RI in 2 (blue) and 4 (red, 4 TARPed) TARPs per CP-AMPAR complex is higher compared with 0 TARPs (white, TARPless). Box-and-whiskers plots meaning as in *Figure 1*. The data from this figure containing statistical tests applied, exact sample number, p values and details of replicates are available in '*Figure 3—source data 1*'.

The online version of this article includes the following source data for figure 3:

**Source data 1.** Polyamine block attenuation of CP-AMPARs.

together with γ2 (*Soto et al., 2014*) (16.58 ± 0.69 pS for 0-TARP *vs.* 24.34 ± 1.69 pS for 4-TARP; p<0.01; n = 10 and 16 respectively; *Figure 4B–C*). Surprisingly, the 2-TARP condition (co-transfection of GluA1 and GluA1:γ2) did not increase single-channel conductance (16.58 ± 0.69 pS for 0-TARP *vs.* 17.03 ± 2.06 pS for 2-TARP; p>0.05; n = 10 for both conditions), indicating that 2 γ2 molecules were not sufficient to increase AMPAR conductance.

While it is true that TARPs have a profound effect on single channel conductance of AMPARs (*Soto et al., 2007*; *Soto et al., 2009*) their effect on the open probability is more controversial.

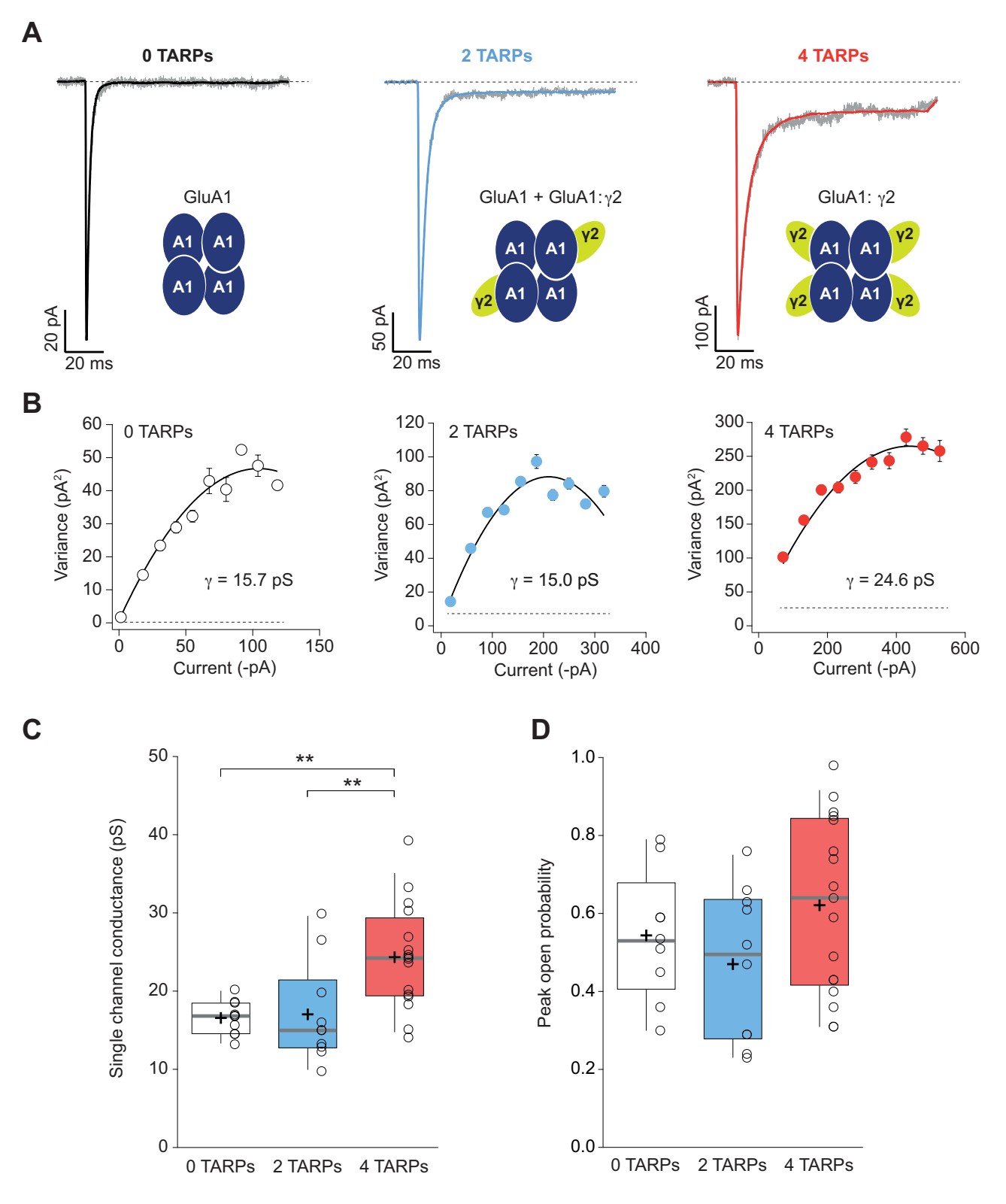

**Figure 4.** Four TARPs are required to increase CP-AMPAR channel conductance. (**A**) Typical responses at a holding potential of −60 mV to rapid application of 10 mM glutamate to excised patches from cells expressing homomeric GluA1 alone (black; average of 84 responses) or together with 2 (blue; average of 91 responses) or 4 (red; average of 223 responses) γ2 subunits. A single trace is shown in gray overlaid with the mean response. (**B**) Current-variance plots for the traces shown in A, the slope of which gave the weighted single-channel conductance. Broken lines show the baseline

*Figure 4 continued on next page*

*Figure 4 continued*

variance and error bars denote SEM. Single channel conductance values for these recordings are presented. (**C**) Pooled data showing an increase of the single channel conductance only in a full-TARPed CP-AMPAR. (**D**) Pooled data for peak open probability of CP-AMPARs. The data from this figure containing statistical tests applied, exact sample number, p values and details of replicates are available in '*Figure 4—source data 1*'.

The online version of this article includes the following source data and figure supplement(s) for figure 4:

**Source data 1.** Channel conductance of CP-AMPARs.
**Figure supplement 1.** Short and long isoforms of GluA4 show the same electrophysiological behaviour.
**Figure supplement 1—source data 1.** Source data.

Some studies have shown an increase of open probability in AMPARs mediated by γ2 (*Suzuki et al., 2008*) while others have not observed such an increase with γ2 nor with other members of the TARP I subfamily (*Soto et al., 2007*; *Shi et al., 2010*). The NSFA allows determination of the number of channels contributing to a given response besides of the unitary conductance. Henceforth, peak open probability ($P_{o,peak}$) can be easily deduced from the experimental mean peak current analysed. We determined the $P_{o,peak}$ of CP-AMPARs when 0, 2 or 4 γ2 were forming part of the complex. As shown in *Figure 4D*, we did not find any increase of $P_{o,peak}$ regardless of the amount of γ2 in the AMPAR complex (0.54 ± 0.06, 0.47 ± 0.06 and 0.62 ± 0.05 for 0-TARPs, 2-TARPs and 4-TARPs. p>0.05 for all comparisons between groups. One-way ANOVA; n = 9,10 and 17 respectively).

## Characterization of GluA4c

In order to study the effect of stoichiometry on heteromeric GluA2-containing CI-AMPARs, we focused on GluA2/GluA4, the putative AMPAR present in CGCs (*Mosbacher et al., 1994*). However, an alternative splicing short isoform of GluA4 – GluA4c – is highly expressed in CGCs (*Gallo et al., 1992*; *Kawahara et al., 2004*) and we thus considered that GluA2/GluA4c would be a better read-out of CGC AMPARs. Consequently, the examination of this specific combination would permit us to study CI-AMPARs and compare later on the results from expression systems with data extracted from CGCs.

We first explored the behaviour of homomeric GluA4c(flip) receptors – referred hereafter as GluA4c – given the relative scarce data of GluA4c in the literature. When long and short forms of GluA4 were compared we observed that GluA4c homomeric AMPARs were functionally identical to GluA4 homomers. No differences were spotted in the single channel conductance or peak open probability (*Figure 4—figure supplement 1A–B*) measured by means of NSFA. Likewise, desensitization kinetics did not differ between both isoforms (*Figure 4—figure supplement 1C*). Finally, both homomeric receptors presented the same degree of intracellular block by spermine (*Figure 4—figure supplement 1D–E*).

## CI-AMPAR channel conductance is differentially affected depending on γ2 location within AMPAR complex

Once GluA4c was validated, we created a GluA4c:γ2 fusion protein to study AMPAR/TARP stoichiometry in CI-AMPARs. Unlike homomeric GluA1 receptors, in a heteromeric receptor such as GluA2/GluA4c, a 2-TARPed configuration might be achieved by locating the TARPs either in the GluA2 or in GluA4c subunit. Since this might be relevant, we co-transfected GluA2, GluA4c, GluA2:γ2 or GluA4c:γ2 to obtain 0-TARPed, 2-TARPed in GluA4c, 2-TARPed in GluA2 or a fully TARPed heteromeric AMPARs. We studied responses in out-side out patches where GluA2 presence into the AMPAR was evaluated with the linearity of the responses (*Figure 5A*).

NSFA performed on glutamate-evoked responses of patches from tsA201 cells transfected with different combinations (*Figure 5B–C*) depicted a remarkable effect not observed with CP-AMPARs. Although any number of γ2 molecules into the CI-AMPAR was sufficient to increase single channel conductance as reported previously (*Jackson et al., 2011*) (p<0.01 for all comparisons between 0-TARPs and the other TARPed groups; *Figure 5C–D*), the γ2 location within the complex was important in determining the extent of conductance increase. Specifically, γ2 intensified its effect on single channel conductance when it was attached to GluA2 subunit compared with GluA4c (15.85 ± 1.62 pS *vs.* 9.72 ± 1.09 pS for γ2 attached to GluA2 or GluA4c respectively; p<0.01; n = 10 and 20 patches), which represents a 309% and an 189% conductance increase respect to GluA2/GluA4c

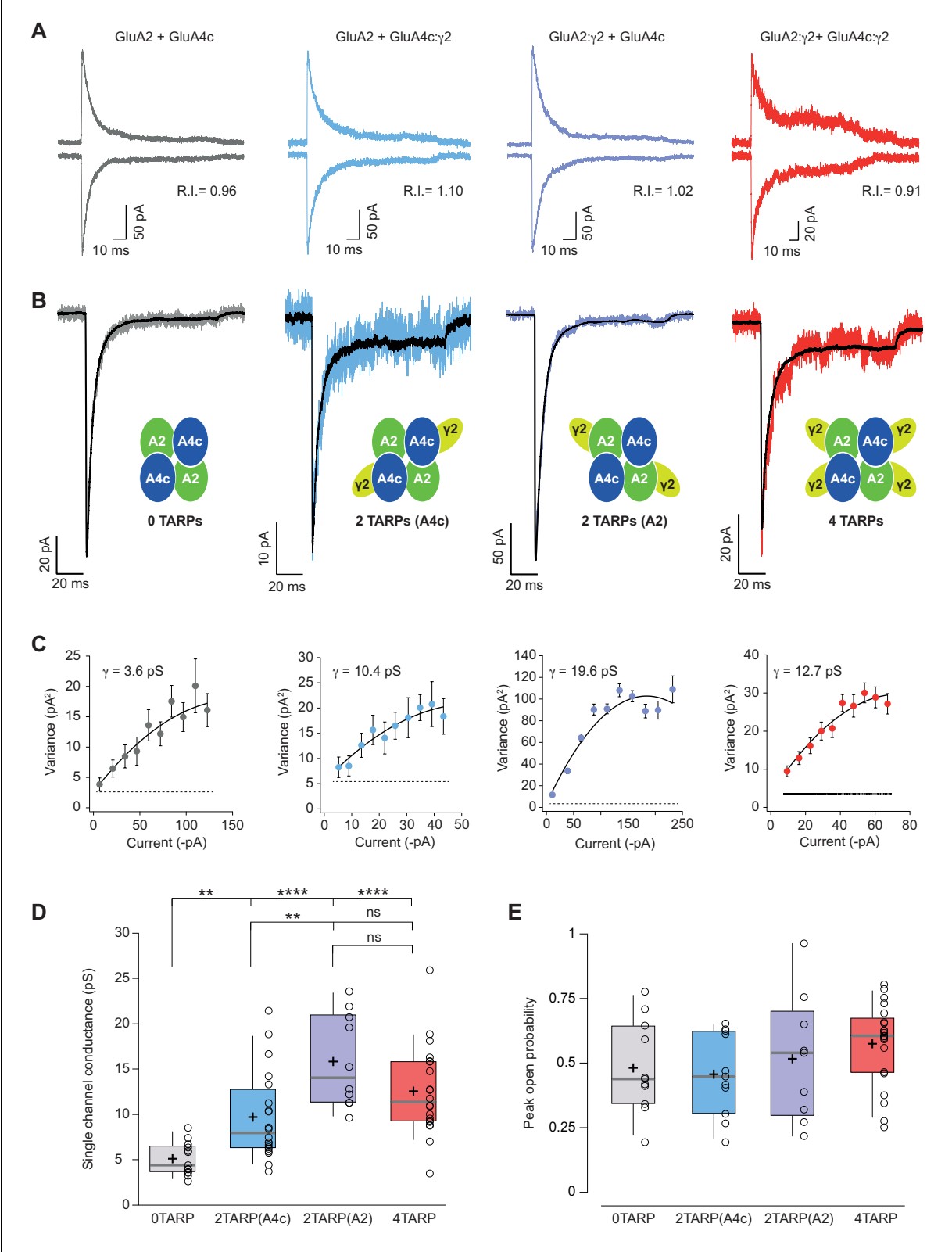

**Figure 5.** Single channel conductance of CI-AMPARs is modulated differently by TARPs depending on their location within the complex. (**A**) Evoked currents by rapid application of 10 mM glutamate from membrane patches at +60 mV (upward traces) and −60 mV (downward traces) with their corresponding RI. The experimental conditions are designated as grey for 0 TARPs per AMPAR, blue for 2 TARPs per AMPAR with γ2 linked to GluA4c, purple for 2 TARPs per AMPAR with γ2 linked to GluA2 and red for 4 TARPs per AMPAR. (**B**) Average traces of current responses evoked at −60 mV

*Figure 5 continued on next page*

*Figure 5 continued*
used for NSNA shown in black overlaid with a representative single response. Insets show the studied combination. (C) Current-variance plots for the recordings shown in B, with the weighted single-channel conductance for the single recordings. (D) Pooled data showing a distinct degree in single channel conductance increase when γ2 is present into the AMPAR complex. (E) Pooled data for peak open probability of CI-AMPARs, where no effect of TARP stoichiometry was evident. The data from this figure containing statistical tests applied, exact sample number, p values and details of replicates are available in '*Figure 5—source data 1*'.
The online version of this article includes the following source data for figure 5:

**Source data 1.** KInetic properties of CI-AMPARs.

(5.13 ± 0.50 pS). Finally, peak open probability deduced from the same NSFA did not seem to be affected regardless of the number of TARP molecules or their location on GluA2-containing heteromers (*Figure 5E*).

## TARP γ2 linked to GluA4c slows down desensitization kinetics of CI-AMPARs

The striking observation about a differential modulation of γ2 depending on its particular position into the receptor prompted us to investigate whether other properties of CI-AMPARs were also affected differentially when γ2 was linked to a specific subunit. *Figure 6A–B* shows that desensitization kinetics of GluA2/GluA4c heteromeric combination (0T) were slowed down only when γ2 was linked to GluA4c subunit irrespective of a 2- or 4-TARP stoichiometry. Indeed, the kinetics of GluA2/GluA4c:γ2 – 2T(A4c) – were not different from GluA2:γ2/GluA4c:γ2–4T – (7.23 ± 0.43 ms for 2T(A4c) *vs.* 8.43 ± 0.61 ms for 4T; p>0.05; n = 21 and 20 respectively; *Figure 6C*) while both combinations were slower than 0T or 2T(A2) (p<0.05; *Figure 6C*). Finally, γ2 linked to GluA2 subunits did not seem to affect GluA2/GluA4c desensitization (4.76 ± 0.28 ms for 0T *vs.* 5.42 ± 0.40 ms for 2T(A2); p>0.05; n = 14 and 11, respectively). Thus, kinetic behaviour in CI-AMPARs is apparently not changed by γ2 unless the TARP is attached to GluA4c subunit. Concerning the activation time to reach the peak current (rise time), we did not notice any variation amongst the different combinations tested (*Figure 6D*).

## TARP γ2 hinders recovery from desensitization of CI-AMPARs by acting on specific subunits

We also checked how recovery from desensitization of GluA2/GluA4c heteromers was affected by distinct TARP stoichiometries. Previous reports have shown that the recovery of GluA2 is unaffected by γ2 (*Cais et al., 2014*). In line with this work, we did not observe faster recoveries regardless of the amount of TARP present into the AMPAR, contrary to what happened in CP-AMPARs. Interestingly, we found that recovery from desensitization was significantly slowed in the 2T(A2) receptor (*Figure 6E–G*).

## CI-AMPAR pharmacology is altered depending on their TARP stoichiometry

A previous study revealed that AMPAR efficiency to the partial agonist kainate is enhanced as the number of TARPs in the AMPAR complex increase (*Shi et al., 2009*). We wondered whether perampanel, a non-competitive inhibitor of AMPARs, would vary its blocking effect depending on the number of TARPs present on AMPARs. To address this question, we performed recordings where we rapidly applied perampanel in a set of experiments where whole-cell currents were activated in transfected tsA201 held at −60 mV with 100 μM AMPA plus 50 μM cyclothiazide to avoid desensitization (*Figure 6—figure supplement 1A*). The outcome of those experiments did show the same pattern as the one found for desensitization: TARP γ2 modified the percentage of block only when attached to GluA4c subunit. CI-AMPARs with 2 TARPs at the GluA2 subunit displayed a similar block as TARPless CI-AMPARs (47.33 ± 5.95% for 0T *vs.* 46.13 ± 5.47% for 2T(A2); p>0.05; n = 6 and 5, respectively; *Figure 6—figure supplement 1C*). However, the block by perampanel in a 2-TARPed at GluA4c (70.65 ± 8.06; n = 7) and in a 4-TARPed CI-AMPARs (71.64 ± 6.76; n = 6) was higher than for TARPless receptors (p<0.05 for 0T *vs.* 2T(A4c) and 0T *vs.* 4T).

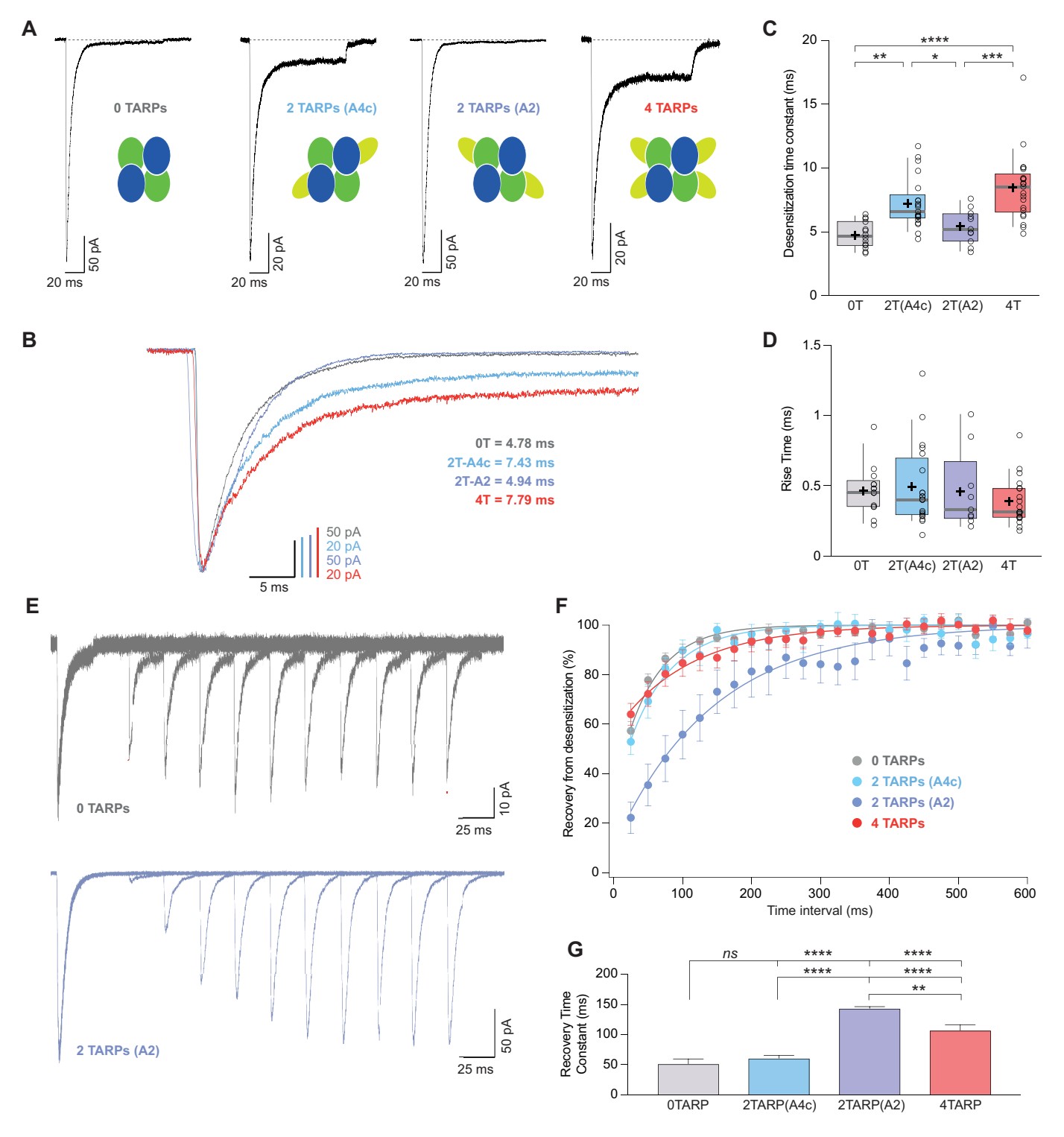

**Figure 6.** CI-AMPAR kinetics differ upon γ2 attachment to GluA4c or GluA2 subunit. (**A**) Representative traces of currents at −60 mV from cells expressing CI-AMPARs without or with TARP γ2 linked to GluA subunits. Under the traces a scheme of the subunits forming the receptors with γ2 associated to different AMPAR subunits is shown. (**B**) Peak-scaled normalization from traces shown in A for a better comparison of desensitization kinetics. (**C**) Weighted time constant of desensitization ($\tau_{w,des}$) where is clear that desensitization is slowed only when γ2 is linked to GluA4c subunit. (**D**) Rise time of the current activation is not changed by the AMPAR-TARP stoichiometry. (**E**) Representative traces monitoring recovery from desensitization for CI-AMPAR in cells expressing 0 TARPs or 2 TARPs linked to GluA2 subunit where it is manifest the difference between the two

*Figure 6 continued on next page*

*Figure 6 continued*

conditions. (F) Recovery of desensitization kinetics showing a relatively slow recovery only in 2-TARPed (located in GluA2) CI-AMPARs. (G) Recovery time constant values for the experiments showed in E and F. The data from this figure containing statistical tests applied, exact sample number, p values and details of replicates are available in '*Figure 6—source data 1*'.

The online version of this article includes the following source data and figure supplement(s) for figure 6:

**Source data 1.** Channel conductance of CI-AMPARs.
**Figure supplement 1.** The block of the non-competitive antagonist perampanel varies with AMPAR-TARP stoichiometry.
**Figure supplement 1—source data 1.** Source data.
**Figure supplement 2.** Effect of C549L and C550L AMPAR mutations in AMPAR-TARP modulation.
**Figure supplement 2—source data 1.** Source data.

## CL mutation differentially blunts γ2 function in CP- and CI-AMPARs

We next decided to inquire into the differential effect of γ2 fused to either GluA2 (conductance) or GluA4c (kinetics) in CI-AMPARs. Specifically, we wanted to ascertain if γ2 was acting on the subunit that it was fused to or, on the contrary, it was operating on the non-fused subunit. To address this question, we decided to selectively blunt γ2 function on either GluA2 or GluA4c by using the CL mutation (*Hawken et al., 2017*). This point mutation in the first TM domain of AMPAR subunit does not interfere with the binding of both proteins but prevents γ2 from modulating the kinetics of AMPARs.

First, we validated the CL mutation in a fully TARPed homomeric CP-AMPAR. When we compared responses from cells transfected with either GluA4c:γ2 or GluA4c(C550L):γ2, we confirmed that the slowing of desensitization kinetics caused by γ2 was prevented in the C550L mutated receptor (6.31 ± 0.80 ms for GluA4c:γ2 *vs.* 3.60 ± 0.60 ms for GluA4c(C550L):γ2; p=0.036; student's t-test; n = 11 and 6, respectively; *Figure 6—figure supplement 2A*) as previously reported for a CP-AMPAR (*Hawken et al., 2017*). Moreover, we expanded the previously described CL mutation blunting effects by observing that it had also a dampening effect on single channel conductance modulation, avoiding the typical increase induced by γ2 (*Soto et al., 2007*) (23.84 ± 1.87 pS for GluA4c:γ2 vs. 17.46 ± 2.99 pS for GluA4c(C550L):γ2; p=0.031; student's t-test; n = 10 and 6, respectively; *Figure 6—figure supplement 2B*).

Then, we explored the CL mutation on GluA2/GluA4c CI-AMPARs. The blunting C550L mutation did not prevent the slowing of kinetics caused by γ2 attached to GluA4c (p=0.9189; student's t-test; n = 21 and 12, respectively; *Figure 6—figure supplement 2D*), However, the milder increase in conductance caused by γ2 when it is attached to GluA4c (9.72 ± 1.09 pS – 2T(A4c)) was avoided and conductance levels were very similar to the ones seen in TARPless condition (6.03 ± 0.63 pS in 2T (A4c(550L)) *vs.* 5.15 ± 0.50 pS in 0T; p=0.2775; *Figure 6—figure supplement 2F*). Surprisingly, the C549L mutation on GluA2 did not affect the increase in conductance caused by γ2 when it is linked to GluA2 (p=0.8556; student's t-test; n = 10 and 6, respectively; *Figure 6—figure supplement 2E*). Moreover, no effect on CL mutation was observed in 4T condition with both subunits mutated (data not shown). The results denote a complex situation in which the CL mutation does not have a clear effect as occurs with homomeric AMPARs.

## Somatic AMPARs from cerebellar granule cells display features of 2-TARPed AMPARs

CGCs have a high expression of GluA2 and GluA4c AMPAR subunits. Besides, these neurons only express two TARPs (γ2 and γ7) (*Fukaya et al., 2005*) and no other auxiliary subunits such as cornichons have been described to be functionally present. While γ2 has been proved to be essential in AMPAR signalling in this cell type (*Chen et al., 2000*), the role of γ7 does not seem to be important to determine CI-AMPAR expression in granule cells (*Studniarczyk et al., 2013*). This converts CGCs into a well-defined system to study AMPARs. Thus, to determine whether CI-AMPARs in CGCs showed properties indicative of a given TARP stoichiometry, we firstly extracted somatic patches from 6 to 8 days in vitro CGC cultures (*Figure 7A*) and applied the selective agonist AMPA at 100 µM. We obtained non-rectifying responses such as the one shown in *Figure 7B* and performed NSFA (*Figure 7C–D*) to extract single channel conductance and peak open probability values from the recorded responses (*Figure 7E–F*; orange boxes), and we also calculated desensitization values

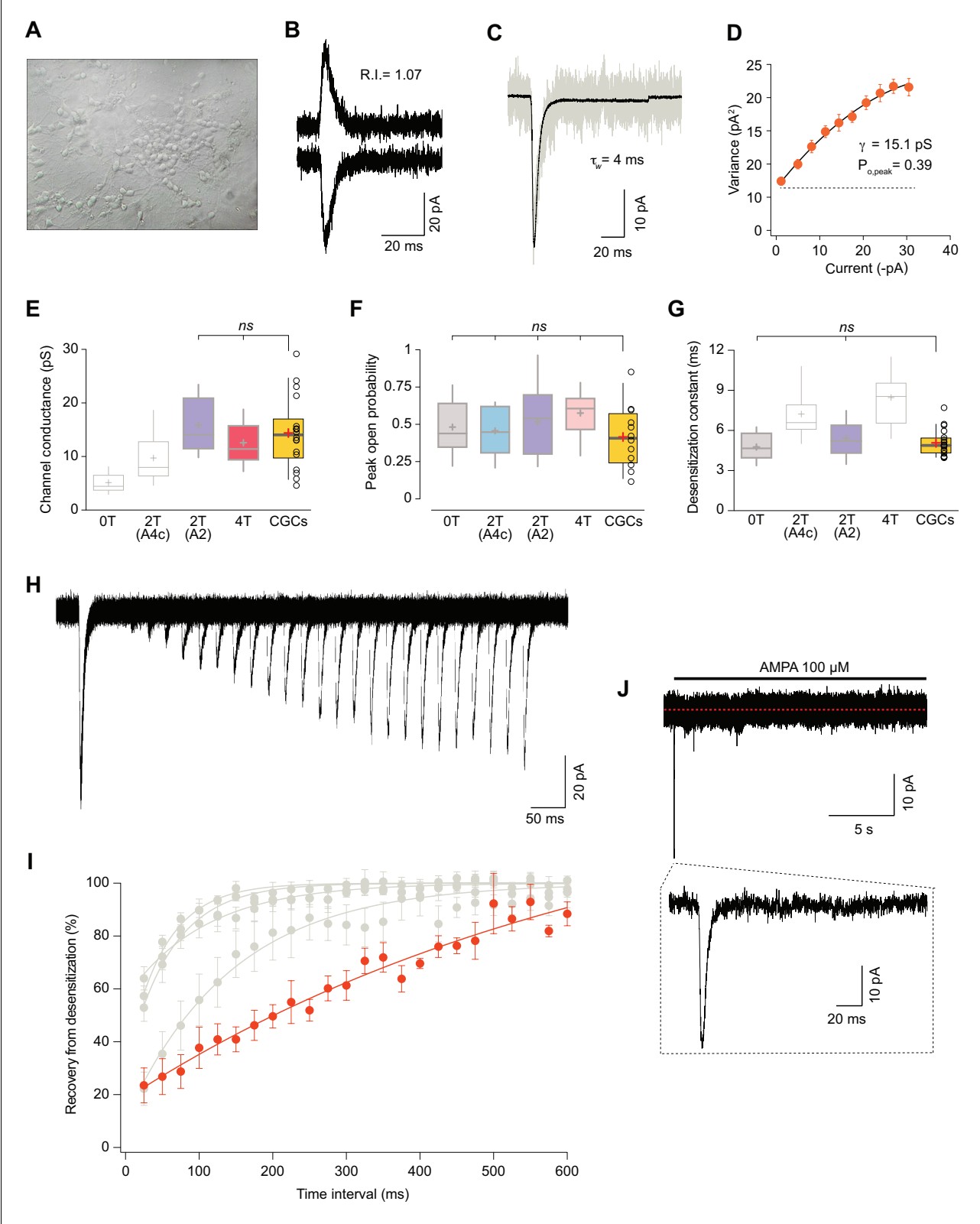

**Figure 7.** Somatic currents from CGCs exhibit properties of GluA2:γ2 + GluA4c CI-AMPARs. (**A**) CGCs in culture after 7 days in vitro. (**B**) Traces at +60 mV and −60 mV evoked with 100 µM AMPA from a CGC somatic patch showing the typical lineal response of a CI-AMPAR. (**C**) Representative response of current evoked at −60 mV by rapid application of 100 µM AMPA to somatic patches from CGCs. Grey: representative single response; black: average of 275 stable responses. (**D**) NSFA from the recording in C. (**E**) Data showing comparison of single channel conductance values obtained

*Figure 7 continued on next page*

*Figure 7 continued*

in CGCs (orange box) with recordings from transfected cell lines. The conductance values obtained resembled (without significant difference) to the ones seen with 2T(A2) or 4T conditions (marked in bold grey). (F) Comparison of peak open probability values from CGCs were no significant differences from recordings in cell lines was observed. (G) Compared data of desensitization time constant (ms) from CGCs with recordings in tsA201 cells. The results are no significantly different from conditions with 0T or 2T(A2). (H) Representative trace from two-pulse protocol monitoring recovery from desensitization for CGCs somatic patches to 100 µM AMPA application. (I) Recovery from desensitization kinetics of CGS somatic AMPARs compared with recoveries of GluA2:GluA4c combinations shown in Figure 6F (in grey). (J) Representative response to a 100 µM AMPA application for 20 s in a somatic patch from CGCs to test for the presence of γ7. No re-sensitization of the receptors is observed in the trace. Inset: magnification of 200 ms showing the initial fast desensitizing response. The data from this figure containing statistical tests applied, exact sample number, p values and details of replicates are available in '*Figure 7—source data 1*'.

The online version of this article includes the following source data and figure supplement(s) for figure 7:

**Source data 1.** Cerebellar granule cells properties.
**Figure supplement 1.** Recovery of desensitization kinetics showing no differences between agonists used to evoke currents in 2T(A2) condition.
**Figure supplement 1—source data 1.** Source data.

from those same patches (*Figure 7G*; orange box). We next compared those results with the ones obtained in expression systems shown previously in *Figures 5* and *6*. *Figure 7E to G* show, for each analysed parameter, the CGCs results (orange box) together with the different combinations analysed in which dull colours represent the comparable combinations (with no significant differences) and in light grey the improbable combination responsible for currents found in CGCs (p<0.05). Taken together, it was clear that values obtained from CGCs were closely equivalent to values from recombinant receptors only for a 2-TARPed CI-AMPAR with the two TARP molecules linked to the GluA2 subunit.

We next studied the recovery from desensitization in CGCs and compared the results with the different CI-AMPAR combinations analysed in this study. Surprisingly, we consistently recorded currents with an exceptionally slow recovery that were different from any CI-AMPAR studied (*Figure 7I*). Given that no other TARP seems to be present on CGCs except γ7, we wondered if this auxiliary subunit was responsible for the extraordinary slow recovery seen on AMPARs from CGCs. We examined the presence/absence of γ7 by applying a 20 s jump of AMPA to CGC somatic patches. This type II TARP has been demonstrated to confer a remarkable feature to either homomeric and heteromeric GluA2-containing AMPARs consisting in resensitization of the current a few seconds after desensitization (*Kato et al., 2007*; *Kato et al., 2010*). We did not appreciate any change in desensitized current in the presence of the agonist for the whole recording period (*Figure 7J*) ruling out any contribution of γ7 to CI-AMPAR currents from somatic CGCs. We also speculated about the possibility that the use of the selective agonist AMPA to activate the currents might slow down the recovery from desensitization since AMPA is known to slow this parameter (*Zhang et al., 2006*). However, recordings from cell lines expressing GluA2:γ2+GluA4c did not vary when we used AMPA as agonist (*Figure 7—figure supplement 1*), ruling out this possibility.

All together, these results seem to indicate that CI-AMPARs from somatic CGCs are very possibly modulated by two γ2 in a similar manner as in our 2T(A2) condition.

## Discussion

One of the first evidences of a functionally variable AMPAR-TARP stoichiometry came from the observation that mEPSCs were differentially altered depending on the TARP expression levels (*Milstein et al., 2007*). Shortly after, by using fusion constructs similar to the ones used in our study it was seen that the pharmacology – the kainate efficacy – of recombinant AMPARs was stoichiometry-dependent (*Shi et al., 2009*). This same work suggested that AMPARs from hippocampal pyramidal and dentate gyrus granule neurons were 4 and 2-TARPed, respectively. Finally, a recent work has provided evidence for the presence of different stoichiometries in cerebellar cells – 2 and 4 TARPed AMPARs in stellate and Purkinje cells, respectively (*Dawe et al., 2019*). We have expanded these previous findings by carefully dissecting the effect of different stoichiometries on basic AMPAR properties. We demonstrate a sophisticated modulation of TARPs either in CP- and CI-AMPARs and we propose that 2 TARPs attached on the gating-controlling BD 'pore distal' subunit GluA2 modulate somatic AMPARs in CGCs in line with recent reports (*Herguedas et al., 2019*).

## Graded *vs.* all-or-nothing modulation of CP-AMPARs by γ2

We have found a clear stoichiometry dependence for some of the parameters modulated by γ2 (i.e: desensitization kinetics or attenuation of polyamine block). However, a more intricate modulation appears to be present since a full TARPed GluA1 receptor is necessary to vary other CP-AMPAR intrinsic properties such as single channel conductance. On the other hand, other characteristics are not altered by the number of TARPs acting on AMPARs (open probability or rise time). Thus, a multi-faceted scenario arises regarding how TARPs alter AMPAR behaviour. It might be interesting in the future to test whether other auxiliary subunits from the TARP family modulate the AMPAR features studied here, especially single channel conductance since Ib TARPs (γ4 and γ8) seem to associate with AMPARs in a 2-TARPed dependent manner (*Hastie et al., 2013*; *Herguedas et al., 2019*) although they clearly produce an increase in AMPAR channel conductance (*Shi et al., 2010*; *Suzuki et al., 2008*; *Soto et al., 2009*).

In our hands, rise time of the current in GluA1 appears not to be affected by γ2 molecules. In line with that, for this same homomeric AMPAR it has been shown in HEK cells that only TARPs γ4 and γ8 are able to increase the period of time to the peak response (*Milstein et al., 2007*). Previous mEPSCs recordings from CGCs, which are mediated by CI-AMPARs have revealed no differences in the rise time of these quantal synaptic currents when the 'dose' of TARP γ2 was altered into the receptor by using homozygous or heterozygous stargazer mice (*Milstein et al., 2007*). Accordingly, our data of rise time of GluA2/GluA4c seems to indicate that AMPAR current activation is independent of the number of TARPs modulating the receptor in CI-AMPARs.

In the present study of CP-AMPAR:TARP stoichiometry a caveat arose when the 2-TARPed condition (transfection of GluA1 and GluA1:γ2 plasmids) was investigated since some homomeric receptors (either TARPless or fully TARPed) might be expressed on the plasma membrane. However, there is some evidence from this work in favour of a major heteromeric population when both constructs are co-transfected. The vast majority of the responses when recording GluA1(Q) + GluA1(R) were linear since in those conditions – *that is, in the presence of GluA1(Q)* – it is difficult that GluA1 (R) homomers contribute significantly. In that situation, we observed the same effect in kinetic desensitization for 0 *vs.* 2 TARPed AMPARs as the ones seen with GluA1(Q), which supports the view of a predominant heteromeric population *vs.* two homomeric ones. Moreover, the fact that some intrinsic properties are undoubtedly affected in the 2-TARP condition (i.e. polyamine block attenuation or desensitization) while others are clearly not (channel conductance or rise time), in the same patches argues in favour of the assembling of mainly heteromeric receptors. Finally, it is worth mentioning that in crosslinking experiments we have observed an intermediate molecular weight of surface AMPARs in GluA1 + GluA1:γ2 transfection compared with GluA1 transfection (with a band at a lower weight) or with GluA1:γ2 (with a band at a higher molecular weight) suggesting that the degree of homomeric contamination in a heteromeric condition is minimal (data not shown). Unluckily, we cannot rule out a small presence of a mixed population despite these indications that a major 2-TARPed population exists in the GluA1+GluA1:γ2 condition.

## CI-AMPAR/TARP stoichiometry in granule cells from the cerebellum

Our experiments indicate that, in terms of TARP presence, 2-TARPed rather than 4-TARPed AMPARs are responsible for somatic responses in CGCs. In fact, the overexpression of TARP γ2 in this cell type increased kainate affinity (*Milstein et al., 2007*), indicating that CGCs are not totally saturated by TARPs since the efficiency to kainate is strongly dependent on the number of TARPs in the complex (*Shi et al., 2009*). When comparing the data from GluA2/GluA4c heteromeric receptors using fusion proteins with those from CGCs, it is evident that neither zero TARPs (low conductance) nor four TARPs (slow desensitization kinetics) are modulating somatic AMPARs in CGCs. Importantly, the findings in CGCs closely recapitulated those on expression systems only when 2 TARPs were attached to GluA2 subunit. It has been suggested that TARP subtypes might have different binding sites in the AMPAR complex (*Greger et al., 2017*) on the basis that, for example, only two γ4 can co-assemble with AMPARs as seen with single-molecule photobleaching in live cells (*Hastie et al., 2013*).

Hippocampal CA1 neurons express almost exclusively heteromeric CI-AMPARs (*Wenthold et al., 1996*; *Lu et al., 2009*) and present a 2-TARPs stoichiometry together with 2 CNIHs (*Gill et al., 2011*). This stoichiometry might not be possible in CGCs due to the lack of cornichon homolog

proteins (*Schwenk et al., 2009*). However, in CGCs, γ7 could potentially be playing a role at somatic AMPARs although it has been shown to selectively suppress somatic Cl-AMPARs in CGCs (*Studniarczyk et al., 2013*) and not to have an important involvement in excitatory transmission (*Yamazaki et al., 2015*). The absence of resensitization – a γ7 hallmark (*Kato et al., 2007*) – in CGC somatic patches rules out γ7 functional presence (*Figure 7J*). Recent reports show evidence that this recovery of the current upon prolonged agonist application might be a common feature of all TARPs (*Carbone and Plested, 2016*) including γ2. However, resensitization is a fingerprint that indicates the presence of 4-TARPed AMPARs (*Kato et al., 2010*). Indeed, Purkinje cells from stargazer mice, with low γ7 stoichiometry lack this feature (*Gill et al., 2011*). Thus, the absence of resensitization, a characteristic signature of a fully TARPed receptor, reinforces the view of a 2-TARPed conformation in CGCs.

The results in expression systems show that the 2-TARPed stoichiometry (with γ2 fused to GluA2) displayed the slowest recovery from desensitization (*Figure 6F*) but somatic CGC responses displayed even a slower recovery rate. The use of AMPA as agonist in CGC experiments to circumvent other glutamate-activated receptors – especially kainate receptors present in CGCs (*Bahn et al., 1994*; *Belcher and Howe, 1997*) – might account for the slow recovery since this agonist is known to speed entry into desensitization and to slow recovery, relative to glutamate (*Zhang et al., 2006*). Surprisingly, the use of AMPA as agonist in the 2T(A2) condition did not slow recovery from desensitization compared with glutamate (*Figure 7—figure supplement 1*). One possible explanation could be the presence of γ2 into the AMPAR. In the absence of TARPs, it has been demonstrated that an inverse relationship between the affinity of the agonist and recovery from AMPAR desensitization exist (*Zhang et al., 2006*). As seen in many studies, γ2 causes a drastic reorganization of the complex and a well-known consequence of the presence of γ2 is the increased affinity of the receptor for glutamate (*Priel et al., 2005*; *Tomita et al., 2005*). Therefore, the change in the affinity for glutamate induced by γ2 might potentially explain that recovery from desensitization in our conditions was not changed when stimulating with AMPA or glutamate.

On the other hand, the use of AMPA *vs.* glutamate may also potentially alter the outcome of other properties. When we checked both agonists on GluA4c, neither weighted single-channel conductance nor open probability were changed (data not shown). Similar channel conductance estimates were reported, regardless of the use of AMPA or glutamate as agonist (*Swanson et al., 1997*). Furthermore, for GluA2/GluA4, similar conductance values have been described for both agonists (5.5–6 pS) (*Swanson et al., 1997*), matching the values obtained in our and other studies (*Jackson et al., 2011*) using the agonist glutamate. Conversely, in our hands, AMPAR desensitization kinetics of GluA4c seemed to be significantly slower when AMPA was used as agonist (data not shown) despite previous reports indicating that the kinetic properties of AMPA-activated GluA4 homomers were comparable to those activated by glutamate (*Swanson et al., 1997*). In principle, this might confuse the interpretation when comparing recordings in expression systems (glutamate used as agonist) with neurons (AMPA used as agonist). However, the kinetics of the currents evoked with AMPA in CGCs were fast – indeed as rapid as the quicker responses observed in expression systems with glutamate. Therefore, it would be expected that the AMPAR-mediated responses in CGCs using the agonist glutamate would be even faster than using AMPA as agonist – in any case kinetics would be overestimated – which still rules out that the possible TARP stoichiometry present in CGCs were any of the slow combinations: 4 TARPs or 2 TARPs attached to GluA4c.

The remarkably slow recovery of the currents in CGCs could be also attributed to the presence of CKAMPs in the native AMPAR complex since recovery from desensitization is strongly reduced either in neurons and heteromeric recombinant AMPARs by the members of the CKAMP family (*Farrow et al., 2015*; *von Engelhardt et al., 2010*; *Klaassen et al., 2016*). Notably, one of the members of the family, CKAMP39, is highly expressed in the cerebellum (*Farrow et al., 2015*; *von Engelhardt, 2019*). Considering the reported absence of other members of the TARP family in CGCs, the functional absence of CNIH2 (*Shi et al., 2010*) and the really low expression of other AMPAR auxiliary proteins as GSG1L and CKAMP44 (shisa9) (*Zeisel et al., 2018*), CKAMP39 might be a potential good candidate. Future works will elucidate whether this protein plays a role in AMPARs form CGCs.

## Final remarks

AMPAR responses depend on the subunit composition but also importantly rely on the nature of the auxiliary subunit/s accompanying them. This work adds important information about the extraordinary degree of functional variety in AMPARs and shows that AMPAR properties can be modulated differently depending on the number of TARPs, the type of AMPAR and the specific interactions that these auxiliary subunits set up with a given GluA subunit. The broad number of possible combinations of pore-forming plus auxiliary subunits and stoichiometries that can be achieved on AMPARs generate a profuse diversity in glutamatergic responses in the brain. One of the major challenges scientists of the field will face is to resolve the exact composition of the AMPAR complex at different neurons.

# Materials and methods

## Animals and housing

C57BL/6N wild-type mice were housed in cages with free access to food and water and were maintained under controlled day–night cycles in accordance with the NIH Guide for the Care and Use of Laboratory Animals, the European Union Directive (2010/63/EU), and the Spanish regulations on the protection of animals used for research, following a protocol approved and supervised by the CEEA-UB (Ethical Committee for Animal Research) from University of Barcelona with the license number OB117/16, of which DS is the responsible researcher.

## Cell lines culture and transfection

TsA201 cells have been used in this study. This cell line (also known as HEK293T or 293T) is an important variant derived from HEK293 cells containing the SV40 Large temperature sensitive T-antigen (293tsA1609neo), that permits the replication of certain type of constructs containing the viral promoter CMV. This allows for amplification of transfected plasmids and extended temporal expression of the desired gene products in HEK293T and thus to produce increased levels of recombinant proteins compared to HEK293. tsA201 cells were a kind gift of Francisco Ciruela (University of Barcelona), who purchased them from the American Type Culture Collection (ATCC; Reference CRL-3216). The ATCC confirmed the identity of HEK293T by STR profiling (STR Profile; CSF1PO: 11,12; D13S317: 12,14; D16S539: 9,13; D5S818: 8,9; D7S820: 11; TH01: 7, 9.3; TPOX: 11; vWA: 16,19; Amelogenin: X). After the purchase of the cell line, mycoplasma tests were performed in the laboratory on every new defrosted aliquot. Cells were maintained as described in *Gratacòs-Batlle et al., 2014*.

Cells were plated into poly-D-lysine coated coverslips 24 hr before transfection at a density of $1.5 \times 10^6$ cells/coverslips. Cells were then transiently transfected with 0.8–1 µg total cDNA using PEI reagent (1 mg/ml) in a 3:1 ratio (PEI:DNA) according to the manufacturer's directions. The ratio of cDNA used in each condition varied depending on the set of experiments.

## Constructs

GluA1, GluA2 and GluA4 cDNAs (rat, flip isoforms) were old gifts from S. Heinemann (Salk Institute, La Jolla, CA, USA) and P. Seeburg (Max Planck Institute, Heidelberg, Germany).

For this work we used the short version of GluA4, namely GluA4c – first described in 1992 (*Gallo et al., 1992*) – in its flip form. The GluA4c subunit was cloned from mRNA obtained from adult rat cerebellum (*Rattus norvegicus*) cerebellum into a pIRES-mCherry plasmidic vector. The primers used were the following:

Primer Forward (5'−3'): GCGC GCT AGC ATG AGG ATT TGC AGG CAG ATT (GCGC GCT AGC restriction site cloned using NheI-HF enzyme, catalog: NEB #R3131S).

Primer Reverse (5'−3'): CGCGG CTC GAG ATT CTT AAT ACT TTC GGT TCC A (CGCGG CTC GAG restriction site cloned using XhoI-HF enzyme, catalog: NEB #R0146S).

GluA1:γ2 and GluA2:γ2 tandem proteins (into pIRES-GFP vectors) were a generous gift from Ian Coombs (UCL, London, UK). They were obtained as described in *Soto et al., 2014*. GluA4c:γ2 tandem was subcloned into a pIRES-mCherry vector from the GluA4c and the γ2 plasmidic vectors using the same linker region (nine aa linker: GGGGGEFAT). All constructs have been fully sequenced.

### Cerebellar granule cells (CGCs) culture

Primary cultures of CGCs were prepared from pups on postnatal day 7–8 as previously described (*Verdaguer et al., 2002*). The cerebella from 8 to 10 mice pups were collected in 9.5 ml buffer containing 120 mM NaCl, 5 mM KCl, 25 mM HEPES, and 9.1 mM glucose. Thereafter, meninges were carefully removed, and cerebella were dissected out, minced carefully with a blade, and dissociated at 37˚C for 15 min with a solution containing 250 µg /ml trypsin. After 15 min, solution with 2.7 µg/ml DNAse and 8.32 µg/ml soybean trypsin Inhibitor (SBTI) was added. CGCs were separated from non-dissociated tissue by sedimentation and, finally, resuspended in basal medium Eagle's (BME) supplemented with 10% inactivated fetal calf serum, 25 mM KCl, and gentamycin (5 mg/ml), and plated onto poly-L-lysine-coated 24-well plates at a density of 300,000 cells/cm$^2$. After 16–19 hr in culture, cytosine arabinoside was added to a final concentration of 10 µM to inhibit glial cell proliferation. Electrophysiological experiments were performed at 6 to 8 days after platting.

### Generic electrophysiological procedures

Recordings were performed from isolated transfected cells or cerebellar granule cells (CGCs) visualized with an inverted epifluorescence microscope (Axio-Vert.A1; Zeiss). Cells expressing EGFP and/or mCherry fluorescent proteins were selected for patch-clamp recordings. Macroscopic currents were recorded at room temperature (22–25˚C) from outside-out membrane patches or from isolated whole cells using an Axopatch 200B amplifier and acquired using a Digidata 1440A interface board and pClamp 10 software (Molecular Devices Corporation, Sunnyvale, CA).

For all recordings the extracellular solution contained (in mM): 145 NaCl, 2.5 KCl, 1 CaCl$_2$, 1 MgCl$_2$, 10 glucose and 10 HEPES (pH to 7.42 with NaOH). The extracellular control solution applied with the fast agonist application tool was composed by extracellular solution diluted 4% with milli-Q H$_2$O. The extracellular agonist solution applied with the fast agonist application tool was extracellular solution plus 2.5 mg/ml of sucrose with 10 mM glutamate for tsA201 recordings or 100 µM AMPA for tsA201 cells and CGC recordings. The intracellular pipette solution contained (in mM): 145 CsCl, 2.5 NaCl, 1 Cs-EGTA, 4 MgATP, and 10 HEPES (pH to 7.2 with CsOH). The polyamine spermine tetrahydrochloride (Sigma-Aldrich) was added to intracellular solution at 100 µM in all cases, which has been calculated to yield free concentrations in the physiological range (*Soto et al., 2007*).

Patch pipettes were fabricated from borosilicate glass (1.5 mm o.d. and 0.86 mm i.d.; Harvard Apparatus, Edenbridge, UK) by using a Horizontal puller (Sutter P-97) with several resistances depending on the configuration used.

#### Whole-cell recordings

Whole-cell recordings were made from isolated cells using electrodes with a resistance of 3–5 MΩ, giving a final series resistance of 5–15 MΩ. Voltage was held at −60 mV unless otherwise stated. Currents were low-pass filtered at 2 kHz and digitized at 5 kHz. Receptors were activated by bath application of 100 µM AMPA plus 50 µM cyclothiazide (CTZ) to prevent receptor desensitization.

#### Fast agonist application into outside-out patches

Outside-out patches were obtained using electrodes with a resistance of 5–10 MΩ. Rapid solution switching at the patch was carried out by piezoelectric translation of a theta-barrel application tool made from borosilicate glass (1.5 mm o.d.; Sutter Instruments) mounted on a piezoelectric translator (P-601.30; Physik Instrumente). Control and agonist solutions flowed continuously through the two barrels of the theta glass and solution exchange occurred when movement of the translator was triggered by a voltage step (pClamp). To enable visualization of the solution interface and to allow measurement of the solution exchange 2.5 mg/ml sucrose was added to the agonist solution and the control solution was diluted by 4%. Currents were activated by 10 mM glutamate and were low-pass filtered at 10 kHz and digitized at 50 kHz. At the end of each experiment, the adequacy of the solution exchange was assessed by destroying the patch and measuring liquid-junction current at the open pipette; the 10–90% rise time was always <400 µs.

## Recovery from desensitization

To study AMPAR recovery from desensitization, a two-pulse protocol (20 or 25 ms each) was used in which a first pulse was applied followed by a second pulse at different time intervals (from 20 ms to 720 ms). The paired pulses were separated 1 s to allow full recovery from desensitization. To estimate the percentage of recovery, the magnitude of peak current at the second pulse (P2) was compared with the first one (P1).

## Non-stationary fluctuation analysis (NSFA)

To infer channel properties from macroscopic responses, glutamate (10 mM) was applied onto outside-out patches (100 ms duration at 1 Hz) and the ensemble variance of all successive pairs of current responses was calculated using IGOR Pro 6.06 (Wavemetrics, OR) and NeuroMatic (*Rothman and Silver, 2018*). The single-channel current ($i$), total number of channels (N) and maximum open probability ($P_{o,max}$) were then determined by plotting this ensemble variance ($\sigma^2$) against mean current (I) and fitting with a parabolic function:

$$\sigma^2 = \sigma_{B}^2 + \left( iI - \left( \frac{I^2}{N} \right) \right)$$

where $\sigma_{B}^2$ is the background variance. Along with the expected peak-to-peak variation in the currents due to stochastic channel gating, some responses showed a gradual decrease in peak amplitude (*run-down* of the current). The mean response was calculated from epochs containing 50 to 350 stable responses, which were identified by using a Spearman rank-order correlation test (NeuroMatic). The weighted-mean single-channel conductance was calculated from the single-channel current and the holding potential (being -0.055 V after corrected for liquid-junction potential). $P_{o,peak}$ was estimated by dividing the average peak current by theoretical maximum current ($i$N).

## Kinetics of desensitization

The kinetics of desensitization of glutamate-evoked responses and the kinetics of recovery from desensitization were fitted according to a double-exponential function to calculate the weighted time constant ($\tau_{w,des}$):

$$\tau_{w,des} = \tau_{f} \left( \frac{A_{f}}{A_{f} + A_{s}} \right) + \tau_{s} \left( \frac{A_{s}}{A_{f} + A_{s}} \right)$$

where $A_{f}$ and $\tau_{f}$ are the amplitude and time constant of the fast component of recovery and $A_{s}$ and $\tau_{s}$ are the amplitude and time constant of the slow component.

## Current-voltage relationships

In order to study the degree of spermine block of CP-AMPARs at different membrane potentials we applied 10 mM glutamate onto outside-out membrane patches at different holding potentials (from −80 mV to +80 mV in 20 mV increments) and the peak current was used to construct the current-voltage relationship.

The rectification index (*RI*) was defined as the absolute value of glutamate-evoked current at +60 mV divided by that at −60 mV:

$$RI_{+60mV/-60mV} = \frac{|I_{+60mV}|}{|I_{-60mV}|}$$

## Statistical Analysis

Analysis of current waveforms and curve fitting was performed with IGOR Pro 6.06 (Wavemetrics) using NeuroMatic 2.03 (*Rothman and Silver, 2018*; http://www.neuromatic.thinkrandom.com). Statistical analysis was performed using GraphPad Prism version 8.0.1 for Mac OS X (GraphPad Software, San Diego California USA, www.graphpad.com). Comparisons between two groups were performed using the parametric Student' *t*-test for data following a normal distribution or using the non-parametric Mann-Whitney U test for comparisons between groups in which one of them did not follow a normal distribution. Normality of data distribution was tested by Shapiro-Wilk normality test. All statistical differences between three or more groups were examined by one-way ANOVA,

followed by Newman-Keuls multiple comparisons test. P values < 0.05 were considered statistically significant as follows: *p<0.05, **p<0.01, ***p<0.001 and ****p<0.0001.

## Acknowledgements

We would like to thank Jakob Von Engelhardt (Mainz University) for helpful discussion, Francesc Sureda (Universitat Rovira i Virgili) for technical assistance with cerebellar granule cell cultures and Jon Giblin for English revision. This work is supported by Grant BFU2017-83317-P European Union, Fondo Europeo de Desarrollo Regional (FEDER) – Ministerio de Ciencia e Innovación to DS; European Union, Fondo Europeo de Desarrollo Regional (FEDER) - Instituto de Salud Carlos III of Spain Grants FIS PI17/00296 and Retic RD16/0008/0014 to XG, 2017SGR737 (Generalitat de Catalunya) to XG and Maria de Maeztu MDM-2017-0729 to Institut de Neurociencies.

## Additional information

### Funding

| Funder | Grant reference number | Author |
| --- | --- | --- |
| Fondo Europeo de Desarrollo Regional (FEDER) – Ministerio de Ciencia e Innovación | BFU2017-83317-P | David Soto |
| Fondo Europeo de Desarrollo Regional (FEDER) -Instituto de Salud Carlos III | FIS PI17/00296 and RD16/0008/0014 | Xavier Gasull |
| Generalitat de Catalunya | 2017SGR737 | Xavier Gasull |
| Ministerio de Ciencia e Innovación (Spain) | Maria de Maeztu MDM-2017-0729 | Xavier Gasull David Soto |

The funders had no role in study design, data collection and interpretation, or the decision to submit the work for publication.

### Author contributions

Federico Miguez-Cabello, Conceptualization, Data curation, Formal analysis, Investigation, Visualization, Methodology, Writing - original draft, Writing - review and editing; Nuria Sánchez-Fernández, Formal analysis, Investigation, Writing - review and editing; Natalia Yefimenko, Investigation; Xavier Gasull, Conceptualization, Resources, Supervision, Funding acquisition, Writing - review and editing; Esther Gratacòs-Batlle, Conceptualization, Data curation, Supervision, Validation, Investigation, Methodology, Writing - original draft, Writing - review and editing; David Soto, Conceptualization, Resources, Data curation, Formal analysis, Supervision, Funding acquisition, Validation, Investigation, Visualization, Methodology, Writing - original draft, Writing - review and editing

### Author ORCIDs

Federico Miguez-Cabello (ID) https://orcid.org/0000-0002-6154-1922
Xavier Gasull (ID) https://orcid.org/0000-0002-6154-8323
Esther Gratacòs-Batlle (ID) https://orcid.org/0000-0001-8093-3713
David Soto (ID) https://orcid.org/0000-0001-7995-3805

### Ethics

Animal experimentation: The authors state that the animals used in this study were sacrificed following the guidelines of CEEA-UB (Ethical Committee for Animal Research) from University of Barcelona with the license number OB117/16, of which Dr. David Soto is the responsible principal investigator.

### Decision letter and Author response

Decision letter https://doi.org/10.7554/eLife.53946.sa1
Author response https://doi.org/10.7554/eLife.53946.sa2

## Additional files

**Supplementary files**
- Transparent reporting form

### Data availability

All data generated or analyzed during this study are included in the manuscript and supporting files. Source data files have been provided for all Figures.

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
