## [Decision Letter]

**Acceptance summary:**

This paper uncovered how stoichiometry of AMPAR / TARP binding affects AMPAR properties. In particular, the authors demonstrated stoichiometry-dependent modulation of AMPARs by the prototypical TARPγ2, which was differential between calcium-permeable and calcium-impermeable AMPARs. The authors propose that this regulation may be important for cerebellar granule cells, containing two TARPγ2 molecules at somatic AMPARs.

**Decision letter after peer review:**

Thank you for submitting your article "AMPAR/TARP stoichiometry differentially modulates channel properties" for consideration by *eLife*. Your article has been reviewed by three peer reviewers, and the evaluation has been overseen by a Reviewing Editor and Kenton Swartz as the Senior Editor. The following individual involved in review of your submission has agreed to reveal their identity: Yael Stern-Bach (Reviewer #1).

The reviewers have discussed the reviews with one another and the Reviewing Editor has drafted this decision to help you prepare a revised submission.

Summary:

Although regulation of AMPAR properties by the auxiliary TARPs has been extensively studied for many years, the issue of AMPAR/TARP stoichiometry and its impact on receptor function is far from been resolved. Especially, not clear is the native stoichiometry: e.g. is it fixed regardless of the brain region? Is it variable? Is it fixed, but variable between brain regions? Do different TARPs have indeed different stoichiometries? Is TARP stoichiometry affected by other auxiliary proteins? The present manuscript elegantly addresses some of these important questions. First, the authors examine the impact on homomeric calcium permeable (CP) GluA1 receptors and then extend the study to heteromeric calcium impermeable (CP) GluA2/GluA4c receptors, which presumably constitute the major AMPAR form in cerebellar granule cells (CGC). The data presented is of high quality and the conclusions drown are reasonable for most part. However, the presentation of results in some places claims more than the evidence supports.

Essential revisions:

1) The effect of TARP stoichiometry on receptor function has been restricted to TARP γ2. Yet, in the Results section, when making semi-conclusions, the authors frequently refer to TARPs in general without indicating that this may only be valid for TARP γ2. Such reference is also missing in some of the figures. As correctly mentioned in the Discussion section, the situation with other TARPs, especially γ8, may be different (e.g. additive effects versus all-or-none effects). Therefore, in order to avoid wrong interpretations by the naive reader, it should be clear through the manuscript that the analysis was confined to TARP γ2.

2) Analysis of GluA1 homomers: I have some reservations for the argument that the expression of GluA1 + GluA1:γ2 results mostly (or exclusively) in receptors contacting only two γ2 units. The authors rely mostly on the results obtained with the co-expression of GluA1-R looking at the current rectification index – a parameter that is strongly affected by TARP γ2. Therefore, even when analyzing kinetic data from receptors with RI>0.7 they may still record form a mix population. Therefore, the authors cannot rule out recording from a mixture of assemblies. Nonetheless, I agree that the assembly of GluA1 and GluA1:γ2 may not be random as simply expected since the presence of TARP γ2 may affect the stability/instability of certain assemblies and/or the level of surface membrane expression, favoring a symmetric 2:2 assembly. In contrast, for GluA2+GluA4c, there are strong indications in the literature that a 2:2 heteromeric assembly is favored.

3) The authors should tune down the conclusion, put forward in the Abstract, that in CGC a stoichiometry of 2 TARP γ2 exists. The authors rather provide strong evidence for an additional "unknown" factor that may exert not only the extremely low recovery rate from desensitization but also responsible for the effects shared with those observed in the heterologous cells.

4) For better understanding how the AMPAR gating mechanism is regulated by TARPs, given the interesting results obtained for GluA2/GluA4c, it will be interesting to similarly test the effect of TARP γ2 positioning on GluA2/GluA1 heteromers (which comprise the major AMPAR form in the hippocampus). This is especially interesting given the assumption that in both heteromers GluA2 occupies the B/D position. So, since GluA4 and GluA1 are not entirely alike, are the results obtained for GluA2/GluA4c heteromers due to the presence of GluA4c per se or the particular positioning of the TARP γ2? However, this data is not critical for the current manuscript and can constitute a follow-up study.

5) Heteromerization.

5a) "We co-transfected tsA201 cells with GluA1 and/or GluA1:γ2 in such a way that GluA1 homomeric CP AMPARs had zero, two or four TARPs per receptor." In an equal mix of GluA1 +/- TARP fusion, there is nothing to constrain how subunits dimerize. Presumably --, -+, +- and ++ TARP are all equally produced. While on average one might expect two TARPs per intact receptor, the population would nonetheless show appreciable heterogeneity. Receptors can be reliably produced containing zero or four TARPs. It is unclear to me (and unconvincing from the description) how the authors can be confident they have produced receptors containing two TARPs. Their statement needs to be much more cautiously presented (or fully justified, which will be difficult).

5b) Evidence (subsection “CP-AMPAR polyamine block attenuation strongly depends on TARP dosage”, second paragraph) concerning polyamine block and Q/R heteromerization rules out the presence of a large proportion of CP homomers. But it does not negate the point above. Only 1/16th of randomly associated subunits would definitely produce polyamine sensitive receptors (Q homomers). Even such a fully mixed population would not be expected produce rectification stronger than 0.7.

I am concerned in general that the argument in this section seems circular. A contribution from homomeric GluA1R receptors cannot be excluded (in either condition). While the early work from Greger et al. suggested homomeric GluA2Rs are not favoured, recent structural work from the Gouaux lab indicates they do exist in vivo. Furthermore, in the experiments here with GluA1Q:γ2 plus GluA1R, it cannot easily be excluded that heteromeric AMPARs contain a single GluA1Q (and hence a single TARP molecule). Can we be sure that such receptors do not give linear I-V plots? I do not think it is clear from the literature. And if not, one cannot use this to argue the view that two TARPs are present in the GluA2Q experiments.

5c) For GluA2/4c associated with two TARPs on the other hand, subunits should hetero-dimerize in a specific orientation. The difference in properties displayed by the TARP associated GluA2 vs. the TARP associated GluA4c (Figures 5 and 6) are a surprising and potentially important finding.

During biogenesis, TARPs do not normally associate with AMPARs until they are tetramers (Schwenk, 2019). Figures 5 and 6 suggest that tandem TARPs can associate with AMPAR monomers and that this association remains unchanged throughout biogenesis. This raises interesting future questions about the nature of this monomeric interaction, and requires some comment.

6) Recovery from desensitization data.

6a) While overall these data demonstrate some interesting findings, they are not the strongest in design. Figures 2B and 6F reveal that the shortest recovery interval is too long to accurately assess recovery kinetics. In Figure 6F, 50% of "0 TARP" desensitization has already recovered before the first data point.

The effect of TARPs on recovery depends both on the TARP member and the GluA1 subunit: GluA1 recovery is unusually slow and is hastened by γ2 (Priel et al., 2005), whereas the recovery of GluA2 is unaffected by γ2 and is slowed by γ8 (Cais et al., 2014).

Recovery in CGCs: the authors used AMPA as an agonist (presumably to isolate AMPAR responses) but should be aware that this agonist is known to speed entry into desensitization and to slow recovery, relative to glutamate (Zhang et al., 2006). Hence a comparison between HEK cells (using L-glu) and AMPA (CGCs) requires the use of the same agonist.

6b) The display of data in Figure 2C is poor. Whereas all other datasets are displayed as box and whisker plots, for Figure 2C a bar graph with a broken y-axis is used accentuating the apparent change. Presumably this is because the data must be exceptionally tight; whereas the error bars on desensitization kinetic data are ~10%, the errors on the recovery data are less than 1%. Given the issues of accurately assessing the recovery time constant with the protocols used (see point above) I do not think a change of this magnitude can be convincingly shown.

6c) For identification of CGC AMPAR currents, the lack of resensitization/superactivation is discussed at length. However, there is no expectation that these receptors would be saturated with γ7 and they lack γ4 and γ8. Hence this seems a confusing distraction. On the other hand, as the authors themselves point out, the recovery data seem to be a good potential fit with CKAMP39. This is an important point that clearly needs to be examined and tested more rigorously.

7) The authors imply that TARPs in an AMPAR heteromer can 'locate' to a given subunit (e.g. subsection “CI-AMPAR single channel conductance is differentially affected depending on TARP location within AMPAR complex”, first paragraph); this is not the case as a given TARPs engages two neighbouring GluA subunits simultaneously – for example in the GluA1/2 γ8 heteromer, the TARP γ8 binding site is formed by TM1 of GluA1 and TM4 of GluA2, and how these interactions modulate the receptor is currently unclear. It is true however, that the TARP extracellular portion is closer to the GluA2 subunit in this complex.

8) The differential effect of TARP γ2 fused to either GluA2 (conductance) or GluA4 (kinetics) is very interesting but lacks any mechanistic understanding. First, the positioning of the two AMPAR subunits to either the AC or the BD positions, which determine function, is currently unclear in GluA2/4 hets, contrasting with GluA1/2 heteromers where the location of the 2 subunits to the AC vs. BD positions is better understood (He et al., PNAS 2016, Herguedas et al., 2019, Zhao et al., 2019). To address this and advance a mechanistic understanding of their observations the authors could selectively blunt γ2 function on either GluA2 or GluA4 by using the CL mutant in TM1 (Herguedas et al., 2019, see also Hawken et al., 2017). Second, the differential effect of γ2 on either conductance or kinetics could be followed up with mutants in the γ2 extracellular portion (see e.g. Riva et al. *eLife* 2017), to address which part of the TARP is responsible for these functional effects.

---

## [Author Response]

Essential revisions:1) The effect of TARP stoichiometry on receptor function has been restricted to TARP γ2. Yet, in the Results section, when making semi-conclusions, the authors frequently refer to TARPs in general without indicating that this may only be valid for TARP γ2. Such reference is also missing in some of the figures. As correctly mentioned in the Discussion section, the situation with other TARPs, especially γ8, may be different (e.g. additive effects versus all-or-none effects). Therefore, in order to avoid wrong interpretations by the naive reader, it should be clear through the manuscript that the analysis was confined to TARP γ2.

We completely agree with the reviewer, and consequently, we have made the appropriate changes along the manuscript (including the Abstract) referring the results and conclusions to TARP γ2 to avoid any wrong interpretation.

2) Analysis of GluA1 homomers: I have some reservations for the argument that the expression of GluA1 + GluA1:γ2 results mostly (or exclusively) in receptors contacting only two γ2 units. The authors rely mostly on the results obtained with the co-expression of GluA1-R looking at the current rectification index – a parameter that is strongly affected by TARP γ2. Therefore, even when analyzing kinetic data from receptors with RI>0.7 they may still record form a mix population. Therefore, the authors cannot rule out recording from a mixture of assemblies. Nonetheless, I agree that the assembly of GluA1 and GluA1:γ2 may not be random as simply expected since the presence of TARP γ2 may affect the stability/instability of certain assemblies and/or the level of surface membrane expression, favoring a symmetric 2:2 assembly. In contrast, for GluA2+GluA4c, there are strong indications in the literature that a 2:2 heteromeric assembly is favored.

As the reviewer remarks, we cannot rule out the presence of a mix population despite our results suggest that a major 2-f population exists in the GluA1+GluA1:γ2 condition. There are several lines of evidence in favour of a major heteromeric population when GluA1 is transfected together with GluA1:γ2 are:

1) The slowing effect on desensitization in GluA1(Q):γ2 +GluA1(R) experiments. Linear responses extracted from the heteromeric combination (with or without attached γ2) supports the view of a predominant heteromeric population vs. two homomeric ones since RIs would be smaller. Importantly, A1Q:γ2+A1R slowed the kinetics (thus, A1Qγ2 was in the receptor) compared with A1Q+A1R in linear responses (where A1R was present). This implies that the degree of “homomeric” contamination seems to be very low.

2) The fact that some intrinsic properties are affected in the 2-TARP condition (i.e. polyamine block attenuation or desensitization) while others clearly are not (channel conductance, rise time or steady state) strongly argues in favor of the assembling of heteromeric receptors when GluA1 and GluA1:γ2 plasmids are transfected, specially taking into account that the analysis was done from the same patches/recordings.

3) Finally, we have performed some molecular biology experiments to address that issue. (See Annex 1 at the end of this document; we have not included it on the manuscript due to external reasons: the lockdown of the University due to Covid-19 did not allow us to further explore this issue).

In these experiments we have transfected the three conditions (GluA1 alone, GluA1+GluA1:γ2 and GluA1:γ2 alone) in the same way as we did for the electrophysiology experiments and after the crosslinking (with BS^3^) of AMPARs with nearby proteins present at the cell surface, we have investigated the molecular weight of these crosslinked complexes by Western-blot to test whether the presence of γ2 in the AMPAR complex could be observed by a shift in the molecular weight of the whole receptor. The results of these experiments are very indicative.

When we probed the crosslinked complexes for the 3 conditions, we found that the antibodies did not recognize their specific epitopes in all conditions. Specifically, for an unknown reason, anti-GluA1 antibody was not able to recognize GluA1 when linked to γ2 (perhaps by an occlusion of the epitope to be recognized) making it impossible to detect any signal in the 4-TARPed condition. However, we observed an increase of the molecular weight in the 2-TARPed condition compared with 0-TARP condition and importantly, we did not see a band at the level of 2-TARPed molecular weight in the 0-TARP condition. This strongly indicates that when transfecting GluA1+GluA1:γ2, homomeric GluA1 are disfavored. Unfortunately, due to the lack of signal in the 4-TARP condition by using anti-GluA1 antibody, we cannot be sure if this transfection (2TARP) did result in homomeric GluA1:γ2 receptors (4-TARP).

However, when we probed with anti-γ2 antibody after stripping the membrane that we had previously probed with anti-GluA1, we could detect in the 4-TARP lane a band with heavier molecular weight compared with 2-TARP condition and without apparent overlapping in weights in the two conditions. No signal was evident in the 2-TARP lane with anti-γ2 antibody. Again, we are not clear of the reason but maybe the expression was not enough to easily detect any signal. Finally, no signal was detected in the 0-TARP condition lane since γ2 is not present in there.

Although not conclusive, taking these three pieces of evidence together, it seems that in the 2-TARP condition it exists an important and preferential presence of heteromeric receptors. But we agree with the reviewer that some homomeric contamination might be present at low levels. This “homomeric contamination” that we are not able to detect in the crosslinking assays might have some substantial contribution in the more sensitive electrophysiological experiments.

Thus, heeding the reservations of the reviewer, we have clearly indicated along the manuscript this possibility, especially in the Discussion section.

3) The authors should tune down the conclusion, put forward in the Abstract, that in CGC a stoichiometry of 2 TARP γ2 exists. The authors rather provide strong evidence for an additional "unknown" factor that may exert not only the extremely low recovery rate from desensitization but also responsible for the effects shared with those observed in the heterologous cells.

As the reviewer points out, with the data we present, it is certainly not appropriate to indicate that a simple “2-TARP stoichiometry” exists in CGCs. When we referred about a 2-TARP stoichiometry we intended to indicate that CGCs did not have a 4-TARP stoichiometry rather than categorize that the complex contains just 2 TARPs without any other auxiliary protein. We have revised the wording throughout the manuscript to make that idea clearer. At the Abstract level we have changed the last sentence to indicate that just 2 TARPs are present in the complex but this would not be the receptor stoichiometry: “Finally, by comparing data from recombinant receptors with endogenous AMPAR currents from cerebellar granule cells, we have determined a likely presence of two γ2 molecules at somatic receptors in this cell type.”

4) For better understanding how the AMPAR gating mechanism is regulated by TARPs, given the interesting results obtained for GluA2/GluA4c, it will be interesting to similarly test the effect of TARP γ2 positioning on GluA2/GluA1 heteromers (which comprise the major AMPAR form in the hippocampus). This is especially interesting given the assumption that in both heteromers GluA2 occupies the B/D position. So, since GluA4 and GluA1 are not entirely alike, are the results obtained for GluA2/GluA4c heteromers due to the presence of GluA4c per se or the particular positioning of the TARP γ2? However, this data is not critical for the current manuscript and can constitute a follow-up study.

We thank the reviewer for this interesting comment. But as he/she indicates, this might well be a nice follow-up study. We would like to have mentioned this important observation in the Discussion, but this part is quite extensive to add even more information.

5) Heteromerization.5a) "We co-transfected tsA201 cells with GluA1 and/or GluA1:γ2 in such a way that GluA1 homomeric CP AMPARs had zero, two or four TARPs per receptor." In an equal mix of GluA1 +/- TARP fusion, there is nothing to constrain how subunits dimerize. Presumably --, -+, +- and ++ TARP are all equally produced. While on average one might expect two TARPs per intact receptor, the population would nonetheless show appreciable heterogeneity. Receptors can be reliably produced containing zero or four TARPs. It is unclear to me (and unconvincing from the description) how the authors can be confident they have produced receptors containing two TARPs. Their statement needs to be much more cautiously presented (or fully justified, which will be difficult).

The important concern of the reviewer totally concurs with comments of the point 2. As explained above, we have tried to address that issue with crosslinking experiments but even though, we have followed the reviewer’s suggestion to be more cautious and we have mentioned the putative presence of certain degree of heterogeneity in the 2-TARP condition.

5b) Evidence (subsection “CP-AMPAR polyamine block attenuation strongly depends on TARP dosage”, second paragraph) concerning polyamine block and Q/R heteromerization rules out the presence of a large proportion of CP homomers. But it does not negate the point above. Only 1/16th of randomly associated subunits would definitely produce polyamine sensitive receptors (Q homomers). Even such a fully mixed population would not be expected produce rectification stronger than 0.7.I am concerned in general that the argument in this section seems circular. A contribution from homomeric GluA1R receptors cannot be excluded (in either condition). While the early work from Greger et al. suggested homomeric GluA2Rs are not favoured, recent structural work from the Gouaux lab indicates they do exist in vivo. Furthermore, in the experiments here with GluA1Q:γ2 plus GluA1R, it cannot easily be excluded that heteromeric AMPARs contain a single GluA1Q (and hence a single TARP molecule). Can we be sure that such receptors do not give linear I-V plots? I do not think it is clear from the literature. And if not, one cannot use this to argue the view that two TARPs are present in the GluA2Q experiments.

These comments are in line with the concerns in points 2 and 5a.

Although the existence of GluA2R homomers has been demonstrated in vivo by Goaoux lab (Zhao et al., 2019), these homomeric complexes represent a very low percentage of the native AMPARs purified from brain (1.1% according with the publication). This indicates that as previously reported, homomeric GluA2 assembly is very exceptional. Besides, in our experience, their presence in cell lines is highly unfavored. Indeed, when we have performed recordings of homomeric GluA2R in expression systems, we have hardly found any glutamate-evoked response in out-side out patches (seen as a slight deflection of the background noise) and really low current magnitude in the whole-cell configuration. Perhaps the existence of these homomeric GluA2R complexes is favoured in vivo by other proteins that are not present in expression systems. This argues in favour of an insignificant GluA2R or GluA1R population in our experiments.

Despite that, as mentioned before, we have been more cautious when presenting the 2-TARPed condition along the manuscript and we have mentioned the existence of homomeric GluA2 receptors in vivo in the Introduction (first paragraph).

5c) For GluA2/4c associated with two TARPs on the other hand, subunits should hetero-dimerize in a specific orientation. The difference in properties displayed by the TARP associated GluA2 vs. the TARP associated GluA4c (Figures 5 and 6) are a surprising and potentially important finding.During biogenesis, TARPs do not normally associate with AMPARs until they are tetramers (Schwenk, 2019). Figures 5 and 6 suggest that tandem TARPs can associate with AMPAR monomers and that this association remains unchanged throughout biogenesis. This raises interesting future questions about the nature of this monomeric interaction, and requires some comment.

We thank the reviewer for the interesting point raised here about AMPAR biogenesis. The Discussion section would be the appropriate place to include a comment on it. Unfortunately, the Discussion section is too long according with the revisions we received, and even when we have reduced it substantially, it is still pretty long due to new data discussion. So, despite the interest, we have chosen not to add extra information to the manuscript.

6) Recovery from desensitization data.6a) While overall these data demonstrate some interesting findings, they are not the strongest in design. Figures 2B and 6F reveal that the shortest recovery interval is too long to accurately assess recovery kinetics. In Figure 6F, 50% of "0 TARP" desensitization has already recovered before the first data point.

The reviewer is completely right, and we appreciate the comment. In fact, the concern of this point 6a (bad design) together with the one in point 6b (poor display of the data) has prompted us to repeat some of the experiments improving the design and presenting the data as in the rest of the manuscript. Specifically, we have repeated all the experiments for the recovery from desensitization of CP-AMPARs (Figure 2). We have now used a shorter time interval between applications (new experiments: 20 ms interval vs. 50 ms in previous experiments) and we also have increased the number of pulses to allow full recovery of the current. Besides we have determined the recovery time constant in a proper way as we explain in detail below (please, see point 6b). The outcome has been the same as in the previous version, with a graded (previously termed gradual) effect as the number of TARPs is increased into the AMPAR complex.

The effect of TARPs on recovery depends both on the TARP member and the GluA1 subunit: GluA1 recovery is unusually slow and is hastened by γ2 (Priel et al., 2005), whereas the recovery of GluA2 is unaffected by γ2 and is slowed by γ8 (Cais et al., 2014).

We thank the reviewer for this remark, and we have included a comment about GluA2 recovery being not affected by γ2 in the manuscript (in “Recovery from desensitization” from CI-AMPARs).

Recovery in CGCs: the authors used AMPA as an agonist (presumably to isolate AMPAR responses) but should be aware that this agonist is known to speed entry into desensitization and to slow recovery, relative to glutamate (Zhang et al., 2006). Hence a comparison between HEK cells (using L-glu) and AMPA (CGCs) requires the use of the same agonist.

We really appreciate this comment since it is an important point that needed to be tested. As specified by the reviewer, the use of AMPA as agonist slows the recovery of desensitization of AMPARs compared to glutamate (Zhang et al., 2006). Hopefully this could explain the slow recovery in CGCs (where AMPA was used as agonist). We therefore have focused in the condition that is fitting all other parameters studied in CGCs and that might be potentially the functional combination present in these neurons (GluA2:γ2 + GluA4c) and we have studied the recovery from desensitization in this condition using AMPA as agonist. Surprisingly (and unfortunately) we have not seen differences in the recovery using both agonists as we initially were expecting – this data has been included as Figure 7—figure supplement 1).

One possible explanation for this unexpected result could be the presence of γ2 in the AMPAR complex tested. In the paper in Biophysical Journal by Zhang et al., 2006, it is shown that there is a clear inverse relationship between the affinity of the agonist and recovery from desensitization. In that work, though, they studied GluA2(Q) homomers in the absence of auxiliary subunit, unlike our experiments in which γ2 is present. γ2 (and other TARPs) cause a drastic reorganization of the complex as seen in many different studies and a well-known consequence of the presence of γ2 is the increased affinity of the receptor for glutamate (Priel et al., 2005; Tomita et al., 2005). Thus, changes in the structure of the AMPAR complex and/or different changes in the affinity of the receptor for different agonists (AMPA/glutamate) might potentially explain that recovery from desensitization in our experiments was not changed by using these two different agonists. We have discussed this in the Discussion section.

6b) The display of data in Figure 2C is poor. Whereas all other datasets are displayed as box and whisker plots, for Figure 2C a bar graph with a broken y-axis is used accentuating the apparent change. Presumably this is because the data must be exceptionally tight; whereas the error bars on desensitization kinetic data are ~10%, the errors on the recovery data are less than 1%. Given the issues of accurately assessing the recovery time constant with the protocols used (see point above) I do not think a change of this magnitude can be convincingly shown.

We completely agree. The reason for the poor display in Figure 2C is that we previously calculated the recovery time constant value by fitting and exponential equation on the average plot made with all experiments, so we did not have the individual values. We have now made all fittings individually for each single experiment (indeed from new experiments performed – see point 6a of the comments). This way, we have calculated the individual time constant for each cell and we have plotted it as the rest of the bars in the manuscript.

6c) For identification of CGC AMPAR currents, the lack of resensitization/superactivation is discussed at length. However, there is no expectation that these receptors would be saturated with γ7 and they lack γ4 and γ8. Hence this seems a confusing distraction. On the other hand, as the authors themselves point out, the recovery data seem to be a good potential fit with CKAMP39. This is an important point that clearly needs to be examined and tested more rigorously.

It was not really our intention to distract the reader with the resensitization discussion, but to clearly explain the absence of γ7 to form a 4-TARP AMPAR (γ2 and γ7). We have eliminated the γ4/γ8 discussion and significantly shortened this whole part (197 words vs. 407 words in the previous version) in the discussion although we think it is important to briefly mention it.

However, we totally agree with the necessity of performing some additional experiments to confirm or to exclude an effect of CKAMP39 in the possible AMPAR combination present in CGCs. Thus, following the reviewer’s suggestion, we designed a set of experiments to test whether CKAMP39 has a slowering effect on the recovery form desensitization in heteromeric GluA4c/GluA2:γ2 receptors. Specifically we co-transfected GluA4c + GluA2:γ2 with or without CKAMP39 (a plasmid that was a generous gift of Jackob Von Engelhardt from Mainz University) in tsA201 cells. We did record GluA2:γ2+GluA4c finding currents in nearly all patches we obtained. However, we did not find currents in out-side out patches when the mentioned heteromeric combination was transfected with CKAMP39 as well (no current in 15 patches). We also recorded 5 cells in the whole-cell configuration without any measurable glutamate-evoked current. Thus, unfortunately, we did not manage to test if CKAMP39 was the missing factor responsible for the remarkably slow recovery from desensitization found in CGCs.

In that respect, it has been described that CKAMPs diminish the amplitude of AMPAR currents (Farrow et al., 2015). This negative effect has been observed for other members of the family (CKAMP52 and CKAMP59) being this decrease in current amplitude evident with several AMPAR combinations for CKAMP39 (Farrow et al., 2015). In this work by Jakob Von Engelhardt and Yael Stern-Bach groups the diminished currents of AMPARs together with CKAMP39 was attributed to both, reduced expression and trafficking. The ER export machinery clearly differs between tsA201 cells and neurons. Indeed, tsA201 cells are used because they efficiently express exogenous proteins. So, it exists the possibility that CKAMP39 expression in heterologous expression is too efficient and AMPARs are saturated (contrary to neurons where there’s a fine regulation), this way reducing AMPARs to the surface as previously shown. AMPAR trafficking depends on their correct gating at ER level and CKAMP39 has been shown to vary AMPAR gating. AMPARs saturated with CKAMP39 (as putatively might happen in the expression system we are using) might decrease the trafficking precisely because of changes in the gating (especially in a TARPed receptor).

Possibly for these reasons we couldn’t get the specific data necessary to address that specific point. Because of this, much to our regret, we were unable to investigate this issue. Due to length constrains, we have decided not to mention it on the manuscript although perhaps reviewers believe this observation might be of interest for other scientists working in the field and could be included in a revised version.

Future work using knockout mice (not yet available) would elucidate whether apart from two γ2 molecules, CKAMP39 plays a role in AMPARs from CGCs. However, we feel that this question is beyond the scope of this manuscript that focuses on the effect of AMPAR:TARP stoichiometry.

7) The authors imply that TARPs in an AMPAR heteromer can 'locate' to a given subunit (e.g. subsection “CI-AMPAR single channel conductance is differentially affected depending on TARP location within AMPAR complex”, first paragraph); this is not the case as a given TARPs engages two neighbouring GluA subunits simultaneously – for example in the GluA1/2 γ8 heteromer, the TARP γ8 binding site is formed by TM1 of GluA1 and TM4 of GluA2, and how these interactions modulate the receptor is currently unclear. It is true however, that the TARP extracellular portion is closer to the GluA2 subunit in this complex.

The reviewer has raised an interesting point that we did not take into account. Actually, TARPs establish interactions with two neighboring AMPAR subunits at the same time and we did not consider that in the manuscript. We have taken that important information into account along the text making clear that even when γ2 is physically attached to a given subunits and its extracellular loops are in a closer contact with that subunit it can interact with the other subunit. Indeed, we have performed new experiments to elucidate the complexity of the interactions. The goal and outcome of these experiments are detailed in the response to the point number 8.

8) The differential effect of TARP γ2 fused to either GluA2 (conductance) or GluA4 (kinetics) is very interesting but lacks any mechanistic understanding. First, the positioning of the two AMPAR subunits to either the AC or the BD positions, which determine function, is currently unclear in GluA2/4 hets, contrasting with GluA1/2 heteromers where the location of the 2 subunits to the AC vs. BD positions is better understood (He et al., PNAS 2016, Herguedas et al., 2019, Zhao et al., 2019). To address this and advance a mechanistic understanding of their observations the authors could selectively blunt gγ2 function on either GluA2 or GluA4 by using the CL mutant in TM1 (Herguedas et al., 2019, see also Hawken et al., 2017). Second, the differential effect of gγ2 on either conductance or kinetics could be followed up with mutants in the gγ2 extracellular portion (see e.g. Riva et al. eLife 2017), to address which part of the TARP is responsible for these functional effects.

We are very grateful for the suggestion made by the reviewer. Following his/her advice, we performed the CL mutations in the TM1 of AMPAR subunits on GluA4c: γ2 (C550L) and GluA2: γ2 (C549L) plasmids. We hoped that using these mutants that prevent selectively γ2 function over a given subunit without affecting their physical interaction we would be able to determine where was potentially acting the TARP.

We first performed some control experiments in order to validate these mutants. Nakagawa’s lab had demonstrated that CL mutation in GluA2 stabilized the AMPAR-γ2 complex but with a loss of function (no effect of γ2 on slowering AMPAR kinetics; Hawken et al., 2017). Although the effect on the gating modulation had been clearly demonstrated, it is not known if this applies as well to other properties of the receptor, particularly to channel conductance. Thus, we performed a first set of experiments with fully TARPed homomeric GluA4c receptors (either with or without CL mutation). We could recapitulate Hawken’s results in terms of kinetics. Essentially, we observed that in the CL mutants desensitization kinetic of the receptor was not slowed by fused γ2 compared to “wild type” forms as previously reported. Moreover, we observed that the mutation had also a preventive effect on single channel conductance modulation, avoiding the typical increase induced by TARP γ2. These results validated the CL mutants to be tested on GluA2/GluA4c heteromers.

When we performed the CL mutant experiments on heteromeric combinations we found an unexpected lack of effect. Surprisingly, CL mutants did not prevent the kinetic slowering caused by γ2 on GluA4c neither prevented the increase in conductance caused by γ2 attached to GluA2. However, CL mutation abolished the increase in channel conductance elicited by γ2 attached to GluA4c. All together denote a very complex situation totally different to what happens in homomeric AMPARs.

We have included this data as “Figure 6—figure supplement 2” since we believe that adds interesting information for other researchers. Firstly, it shows that CL mutation prevents not only desensitization kinetic effects as reported previously but also channel conductance increase. Secondly, denotes the complexity of TARP modulation on heteromeric combinations and raise new questions whether this lack of CL mutation effect is due to the presence of the edited GluA2 subunit or to the heteromeric receptor ‘per se’.

Regarding the suggestion about using mutants in the extracellular loop of γ2 it is certainly very interesting, but we feel that this might be a nice follow up story. In our modest opinion, the complexity of the mechanistic interactions between TARPs and GluAs (even more complex in a GluA2 heteromeric receptor as we could observe) switch this simple question into a complex issue that we think it should be performed in another study.

Annex 1: Crosslinking assay

BS3 crosslinking assay in tsA201 cells expressing GluA1 alone (0 TARPs), GluA1 andGluA1:γ2 (2 TARPs) or GluA1: γ2 only (4 TARPs). BS^3^(Bis(sulfoccinimidyl) subertae) is a membrane-impermeable bifunctional crosslinker that covalently crosslinks cell-surface expressed receptors to nearby proteins (Bourdeau et al., 2012). When BS^3^ is present (3 lanes on the right), surface tetrameric receptors can be crosslinked to nearby γ2 subunits and we expect to see differences in molecular weight (0 TARP GluA1 tetramer should weight approx. 400 kDa, 2 TARPs tetramer approx. 474 kDa and 4 TARPs approx. 548 kDa) when separating the extracts in denaturing 3-8% tris-acetate gels. The noncrosslinked subunits (intracellular subunits) of GluA1 (100kDa) or GluA1:γ2 (137 kDa) should also be detected. When BS3 is absent (three lanes on the left), there is no crosslinking of surface receptors (no band in high molecular weight regions of the membrane) and all subunits of GluA1 expressed in each condition are denatured and detected as bands in the monomeric form in the western blot.

We first probed the bottom part of the membrane with anti-Actin antibody (blue signal) to confirm that equal protein amounts where loaded in each lane. The upper part of the same membrane was probed with an anti-GluA1 antibody against a C-terminus epitope (red signal). Tetrameric GluA1 surface receptors were detected in different molecular weights in extracts from cells expressing the hypothetical 0 TARP or 2 TARPs condition suggesting that tetramers in the 2 TARPs condition contained γ2 subunits and therefore showed a slightly higher molecular weight than tetramers from the 0 TARP condition. Unfortunately, GluA1:γ2 tetramers were not detected with this antibody probably due to some epitope masking in the tandem form (C-terminus of GluA1 is linked to γ2 and it might produce some conformational change that avoids anti-GluA1 antibody to detect this tandem form). Monomeric/intracellular GluA1 (100 kDa) is also detected with this antibody in 0 TARPs and 2 TARPs extracts, although much less GluA1 seems to be expressed in the 2 TARPs extracts. After stripping the membrane, we probed it with an anti-γ2 antibody recognising an intracellular epitope (green signal). Tetramers were observed in high molecular weight bands only in 4 TARPs condition, and this band had a higher molecular weight than that observed for the tetramers detected with the anti-GluA1 antibody in the 2 TARPs or 0 TARP condition. The intracellular signal corresponding to the monomeric tandem GluA1: γ2 (137 kDa) is observed in the 4 TARPs condition but not in the 2 TARPs condition suggesting that in the latter, GluA1:γ2 has not been correctly expressed. This is intriguing because tetramers detected (in the 2 TARPs condition) with the anti-GluA1 had higher molecular weight than those detected in the 0 TARP condition, and this can only be due to some γ2 present in the tetramer, and it can only come from GluA1:γ2 tandem expression. One explanation for the lack of signal of GluA1:γ2 in the 2 TARPs condition could be that the amount expressed is too low to be detected by the antibody but enough to traffic to the surface with GluA1 subunits. In the crosslinked extracts of 4 TARPs condition a decrease in the intensity of the intracellular band of GluA1:γ2 is probably due to the fast trafficking of this form of receptors.